# Weed Detection Using Deep Learning: A Systematic Literature Review

**DOI:** 10.3390/s23073670

**Published:** 2023-03-31

**Authors:** Nafeesa Yousuf Murad, Tariq Mahmood, Abdur Rahim Mohammad Forkan, Ahsan Morshed, Prem Prakash Jayaraman, Muhammad Shoaib Siddiqui

**Affiliations:** 1Big Data Analytics Laboratory, Department of Computer Science, School of Mathematics and Computer Science, Institute of Business Administration, Karachi 75270, Pakistan; 2School of Science, Computing and Engineering Technologies, Swinburne University of Technology, Melbourne 3122, Australia; 3School of Engineering and Technology, Central Queensland University, Melbourne 3000, Australia; 4Faculty of Computer and Information Systems, Islamic University of Madinah, Medina 42351, Saudi Arabia

**Keywords:** weed detection, deep learning, machine learning, systematic literature review

## Abstract

Weeds are one of the most harmful agricultural pests that have a significant impact on crops. Weeds are responsible for higher production costs due to crop waste and have a significant impact on the global agricultural economy. The importance of this problem has promoted the research community in exploring the use of technology to support farmers in the early detection of weeds. Artificial intelligence (AI) driven image analysis for weed detection and, in particular, machine learning (ML) and deep learning (DL) using images from crop fields have been widely used in the literature for detecting various types of weeds that grow alongside crops. In this paper, we present a systematic literature review (SLR) on current state-of-the-art DL techniques for weed detection. Our SLR identified a rapid growth in research related to weed detection using DL since 2015 and filtered 52 application papers and 8 survey papers for further analysis. The pooled results from these papers yielded 34 unique weed types detection, 16 image processing techniques, and 11 DL algorithms with 19 different variants of CNNs. Moreover, we include a literature survey on popular vanilla ML techniques (e.g., SVM, random forest) that have been widely used prior to the dominance of DL. Our study presents a detailed thematic analysis of ML/DL algorithms used for detecting the weed/crop and provides a unique contribution to the analysis and assessment of the performance of these ML/DL techniques. Our study also details the use of crops associated with weeds, such as sugar beet, which was one of the most commonly used crops in most papers for detecting various types of weeds. It also discusses the modality where RGB was most frequently used. Crop images were frequently captured using robots, drones, and cell phones. It also discusses algorithm accuracy, such as how SVM outperformed all machine learning algorithms in many cases, with the highest accuracy of 99 percent, and how CNN with its variants also performed well with the highest accuracy of 99 percent, with only VGGNet providing the lowest accuracy of 84 percent. Finally, the study will serve as a starting point for researchers who wish to undertake further research in this area.

## 1. Introduction

Crop farming is considered a significant agricultural pursuit for the global economy in the modern era, and over a longer time period, it has had a notable impact on countries’ GDP. In 2018, it contributed 4% to the global GDP and accounts for more than 25% of the GDP for many developing countries. Moreover, with almost 9% of the world population hungry in 2020, agriculture is a powerful source of food, revenue, and employment and is expected to minimize poverty, raise income levels, and boost prosperity for a projected 9.7 billion population by 2050 [1,2]. However, agricultural growth through crop farming is always at risk due to several reoccurring problems, for example, climate change, greenhouse gas emissions, pollution and waste generation, malnutrition, and food wastage [1]. Another serious problem that plagues crop farming is the growth of weeds which leads to significant crop wastage annually. Hence, weed management and removal practices have been adopted for several decades to control weed growth [3,4,5].

Weeds are undesired plants that compete against productive crops for space, light, water, and soil nutrients and propagate themselves either through seeding or rhizomes. They are generally poisonous, produce thorns and burrs, and hamper crop management by contaminating crop harvests. Smaller weed seedlings with a slow growth rate are more difficult to detect and manage than larger ones which grow vigorously. Weed management is complicated because the competitive nature of weeds can vary in different conditions and seasons. For instance, the tall and fast-growing fat hen weed is considered dangerous to adjacent crops, but fat hen seedlings that appear in late summer are considerably smaller in size and not potentially dangerous [6]. Similarly, chickweed is smaller and less dangerous during the summer season, but in winter, it can have a high growth rate and can swamp crops such as onions and spring greens [7,8]. Moreover, weeds can co-exist ‘peacefully’ with the crops earlier on in their growth period but start competing for more natural resources later on. Another difficulty in managing weeds is determining the exact time when a weed actually starts to affect the harvest. Moreover, several weeds, such as couch grass and creeping buttercup, can survive in drought and severe winter weather as they store food in long underground stems. Weeds are also potential hosts for pests and diseases which can easily spread to cultivated crops. For instance, the charlock and shepherd’s purse weeds may carry clubroot and eelworm diseases, while chickweed can host the cucumber mosaic virus [7,9,10]. Finally, different weeds have different seeding frequencies, further complicating weed management; for instance, groundsel can produce 1000 seeds per season, while scentless mayweed might produce 30,000 seeds per plant. These seeds might stay in the soil for decades until exposed to light; for instance, the poppy seed can survive even up to 80 years.

For several decades weeds have been managed, detected, and controlled manually [3,4,5]. The most common method of weed detection is manual surveillance by hiring crop scouts or by tasking crop farmers to do the same, which is expensive, difficult to manage, and infeasible to execute in unfavorable weather conditions Scouts only work on a sample of the field and have to follow a pre-determined randomized pattern (e.g., zigzag). Such a setting does not always ensure that all weeds will be detected and removed. Scouts also carry specialized equipment (e.g., hand-held computers with GPS and geo-tagging), which adds to the expense. They need to repeat the process regularly and fill up a report. All these limitations make crop scouting difficult to manage, and hence, weeds continue to affect crop harvest each year globally.

### Motivation and Contribution

In this paper, we focus on smart farming techniques that can detect weeds in crop images through machine learning methods, particularly DL. This can potentially eliminate the need for crop scouts while scanning the entire field for weeds with no management overload. However, ever since the introduction of Graphical Processing Units (GPUs), DL has demonstrated an unparalleled pace of research and superior performance across a wide variety of complicated applications involving images, text, video, and speech datasets [11,12,13,14,15]. DL was considered nascent till 2010 due to a lack of hardware technology to process its complex architectures. One of the initial researches on weed detection found in 1991 [16], which highlighted the limitations of using tractor-mounted weed detectors, and proposed the use of digital image processing (IP) techniques [17,18] to detect weeds from both aerial and previous manually-snapped photographs by crop scouts. Research efforts using pure IP and CV techniques for automated weed detection remained very limited for the next two odd decades [19,20,21,22]. This paper demonstrates that such applications are still in their infancy with respect to applications in ML and DL [12,23]. Since 2000, researchers have been using ML sometimes in combination with CV to automatically detect weeds from images [24,25,26,27,28,29]. Although most of these works detected weeds with reasonable accuracy, they also highlighted the potential of DL for significantly better performances.

Through some initial searches, we determined that DL applications in research for weed detection have increased considerably since 2015, and they primarily use convolutional neural networks (CNNs) and their variants such as SegNet, GoogLeNet, ResNet, DetectNet, and VGGNet [30,31]. Though some survey articles have been published since 2015 on DL applications for weed detection [32,33], they lack proper SLR.

The rapid pace of research in DL and its potential to provide a competitive performance on complicated image-based recognition tasks motivated us to conduct the SLR on DL applications of weed detection. Our general intent is to extract and summarize the relevant and most recent research and provide concrete future directions regarding industrial applications and academic research. As this domain involves working with image data of weeds, we target applications of CNNs, particularly the most recent and standard published research content for the time period between 2015 and January 2023. In our SLR, we answer the following research questions:**RQ1:** 
What is the trend of employing deep learning to address the problem of weed detection in recent years?**RQ2:** 
Which types of weeds and corresponding crops have been detected using deep learning, and what are the characteristics of the corresponding weed datasets?**RQ3:** 
Which deep learning algorithms are best suited for a particular weed/crop combination?**RQ4:** 
What are the tangible future research directions to achieve further benefit from deep learning applications for weed detection?

RQ1 is addressed in Section 4, where we describe our SLR methodology and analyze the trends and other relevant statistics of extracted papers.

RQ2 and RQ3 are addressed in Section 5. For RQ2, we analyze and identify the datasets that have been used in the papers and summarise the relevant information such as weed types, crop types, and characteristics of weed images (e.g., resolution, size).

To address RQ3, we identify the different DL algorithms in the literature and compare their frequency of usage and performance data. Moreover, we categorize each paper by assigning a unique label based on the usage of ML, IP, and DL, along with characteristics such as training time and performance. We then compare the performance of ML and IP algorithms with DL ones. To further strengthen the analysis, we associate the weed types with the algorithm used and its associated evaluation outcome. Finally, we address RQ4 in Section 8 by analyzing our findings and proposing a set of future research directions to motivate and enhance the DL research weed detection domain.

## 2. Related Surveys

In this section, we discuss in detail the eight articles of literature review which we extracted through our SLR.

In [34], the authors review seven research papers based on deep learning and discuss three previously-used techniques for the classification of weeds such as color-based, threshold-based, and learning-based techniques. The authors review the papers over different parameters such as the type of deep learning used, targeted crops, training setup, the training time of the algorithm, dataset acquisition, dataset strength, and accuracy of the algorithm. Research gaps are also identified, and one of the gaps was the lack of a big dataset which could be a major contribution in this field.

Moreover, in [35], the challenges faced by vision-based plant and weed detection and their solutions have been discussed. Two main challenges of weed detection are the light problems, i.e., the algorithm may work differently due to the presence of light, and discrimination between crop and weed, i.e., sometimes both may look similar. Shading or artificial lighting can be used to control the variation of natural light, or image processing techniques like segmentation of background (and then converting the image into Grayscale) can be used to tackle this problem. For the second problem, different types of IP-based classification techniques were discussed, which were based on shape, texture, height, and DL. The authors discussed the comparison of traditional classification and DL methods. They also highlight the application of online cloud databases as an important future direction to further improve the recognition or detection of weeds and crops.

Furthermore, in [22], the authors summarize different problems and provided solutions to weed classification using IP and DL techniques. Four basic steps of classification, such as pre-processing, image segmentation, feature extraction (biological morphology, spectral feature, visual texture, spatial context), and classification (convolutional machine learning), have been discussed in detail. Some challenges like leaf overlapping, light variation, and stages of plant growth and their solutions were discussed. Semi-supervised learning techniques have been proposed by the authors to improve the current performance of the aforementioned techniques.

In [36], the authors analyze different techniques for weed detection using IoT technology. The authors discuss several DL algorithms employed in the context of IoT and perform their comparative analysis, for example, CNN, SegNet (with a synthetic dataset for achieving higher accuracy), and summarised training set technique with CNet, which is a deep CNN based on image segmentation. The authors also propose an IoT-based architecture where different devices and sensors are connected to one central data server, and users can communicate with the server through the Internet. This model can be controlled by a desktop computer or mobile device. Moreover, in [37], the authors focus on the methods and technologies used in weed detection with particular focus on the requirements of weed detection, its applications, and the system needed for weed detection, such as satellite-based positioning, crop-row following, and multi-spectral images. They have also drawn attention to the limitation of previously constructed detection systems, such as the lack of within-row plant-detection facilities.

In [38], the authors discuss DL techniques and architecture. In the former, they discuss Artificial Neural Networks (ANN), CNN, and Graph Convolutional Networks (GCN), and in the latter, they discuss image classification, object detection, semantic segmentation, and instance segmentation. They also mention the significance of public datasets, specifically carrot-weed, CWF-788, CWF-ID, DeepWeeds, GrassClover, Plant Seedlings, Sugar Beets 2016, Sugar Beet/Weed Dataset, and WeedCorn/Lettuce/Radish, to demonstrate how images were acquired, size of the dataset, pixel-wise annotation and modality. They also discuss data augmentation by mentioning limitations in the size of public datasets to work in varied conditions. They discuss fine-grained learning that overcomes the problem of general deep architectures, which ignores the challenges of similarities between crops and weeds, along with low-rank factorization, quantization, and transferred convolutional filters to solve the resource-consumption problems in analyzing real-time data for weed detection through DL. For the manual collection of datasets for weed, identification could be expensive, so weakly supervised and unsupervised methods can be necessary. For weakly supervised, object detection or segmentation can be used on image-level annotation, and for unsupervised learning, domain adaptation and deep clustering can be used. The existing methods for deep learning cannot deal with new species once a model is trained; to overcome this problem, incremental learning is proposed that is used to extend the existing trained model without retraining it.

Finally, in [39], the importance of reducing the use of herbicides is highlighted, and the authors review current and emerging technologies for this domain in the last 5 years. They classify the discussions into “digital image sensor-based” and “non-digital image sensor-based”. In the former, the shapes and morphological features of weeds are used for detection, and in the latter, reflectance spectra are used to detect weeds. A complete workflow example of weed detection of Romaine lettuce has been discussed. This workflow shows the means of automatic weed detection using deep learning based on YOLO. In all the review papers, the authors did not conduct an SLR, and the focus is apparently to review performance over specific tasks rather than conduct a wide-ranging review of DL applications to weed detection.

Perhaps the paper most related to our work is [40], in which the authors review DL approaches to weed detection based on four steps: data acquisition, dataset preparation, weed detection, and localization and classification of weeds in crops. They develop a taxonomy for DL applications specifying the weed and crop type, the DL architecture applied, and the IP technique. In data acquisition, they detail how data or images have been collected, for example, using digital cameras, public datasets, camera moving vehicles, etc. They discuss and classify 19 public datasets according to several standard parameters, such as modality, dataset size, etc. In the data preparation phase, after acquiring images using different sources, images are prepared for training and testing, which includes different techniques, for instance, image processing, image labeling, image augmentation, etc. Weed detection is classified as a plant-based classification or a weed mapping approach. In the former, every plant needs to be localized in an image before detection, and in the latter, the density of the presence of weed in an image is used to detect that weed. In the last step, the authors discuss different algorithms, such as CNN, YOLO, FCN, GCN, and hybrid models, along with learning methods, such as supervised, unsupervised, and semi-supervised.

Several major differences distinguish our paper from [40]. Firstly, our process of review is more standardized because we conduct an SLR and answer concrete research questions (Section 1 and Section 4). Secondly, we present a more thorough analysis of the SLR results, specifically through analyzing different combinations of algorithms, binning and analyzing individual algorithmic performance, specifying appropriate thematic labels, and analyzing the literature with respect to these labels (Section 7). Thirdly, we use a table to present a comprehensive association of algorithmic performance across different weed types and their respective crops, which provides a strong guideline to analyze current performance in the literature and to determine directions for future research (Section 6). Fourthly, we specify these directions more thoroughly to provide a type of road map for researchers of this domain (Section 8).

## 3. Background Knowledge

Before looking deeper into weed detection techniques, it is necessary first to understand weeds. This section discusses weeds, their various types, and weed detection algorithms.

### 3.1. Weed Types

Weeds can be generally classified as annual, biennial, and perennial [3,4,5]. Annual weeds germinate, bloom, and die within one year, while biennial weeds have a life cycle of two years, with germination and blooming happening in the first year and dying out in the second year. Perennial includes all weeds which last longer than two years in that they can germinate, bloom, and seed for several years. In our 60 extracted papers, authors have used a total of 34 weeds, of which we identified 26 annual and 8 perennial types. We illustrate these weeds in Figure 1, Figure 2, Figure 3, Figure 4, Figure 5, Figure 6 and Figure 7 and discuss them in the following two sections.

For each weed, we label it by its commonly-used published name and mentioned scientific name in parentheses (wherever applicable). We extracted more detailed information about all these weeds from the Invasive Species Compendium section of the Cab Institute’s website [42] and Wikipedia entries [43] along with websites of Garden Organic [44], Crop Protect [45], Gardening Know How [46], Lawn Weeds [47], Farms [48], and the USA Department of Agriculture [49]. Moreover, several weeds are categorized as both annual and perennial, for example, chickweed, but we have considered them as annual weeds for classification purposes.

#### 3.1.1. Annual Weeds

In this section, we list the twenty-six (26) annual weeds used in our extracted papers. (1) Chickweed (*Stellaria media*), (2) Loose silky-bent (*Apera spica-venti*), (3) Velvetleaf (*Abutilon theophrasti*), (4) Shepherd’s purse (*Capsella bursa-pastoris*), (5) Cleaver (*Galium aparine*), (6) Black nightshade (*Solanam nigrum*), (7) Blackgrass (*Alopecurus myosuroides*), (8) Littleseed canarygrass (*Phalaris minor*), (9) Crowfoot grass (*Dactyloctenium aegyptium*), (10) Jungle rice (*Echinochloa colona*), (11) Mayweed (*chamomile*), (12) Fat-hen (*Chenopodium album*), (13) Pigweed (*Amaranthus albus*), (14) Chinee apple (*Ziziphus mauritiana*), (15) Snakeweed (*Gutierrezia sarothrae*), (16) Indian jointvetch (*Aeschynomene indica*), (17) Fescue grass, (18) Bluegrass (*Poa annua*), (19) Meadow grass (*Poa trivialis*), (20) Hare’s ear mustard (*Conringia orientalis*), (21) Turnip weed (*Rapistrum rugosum*), (22) Cocklebur (*Xanthium strumarium*), (23) Field pansy (*Viola rafinesquii*), (24) Japanese hop (*Humulus scandens*), (25) Dicot, and (26) Grass weed (*Monocot*). Collectively, the annual weeds can attack livestock and diverse cereal and vegetable crops, notably wheat, maize, sugar beet, tomato, cotton, rice, carrots, potato, peanuts, and corn. They also grow in different land types like crop fields, pastures, orchards, home lawns, grasslands, arable lands, roadsides, seashores, and wooded areas and in different soil types (sandy, dirty, wet). They pose severe management challenges: they can weaken the crop by more than 70% in some severe cases and can continue to germinate in the soil for decades, along with causing skin diseases to farmers and affecting the milk taste of livestock [6,50,51,52].

#### 3.1.2. Perennial Weeds

The following eight (8) perennial weeds have been used in our extracted papers for weed detection: (1) Canadian thistle (*Cirsium arvense*), (2) Paragrass (*urochloa mutica*), (3) Nutsedge (*cyperus rotundus*), (4) Dockleaf (*Rumex obtusifolius*), (5) Benghal dayflower (*tropical spiderwort*), (6) Sedge weed, (7) Lantana (*Lantana camara*) [53], and (8) Hedge bindweed (*morning glory*). Collectively, like the annual weeds, these weeds can cause severe damage to crop fields, gardens, lawns, and other land types and can survive for several decades in diverse soil conditions [54,55,56,57]. They can grow in water and on profound soils in non-muddy areas and can attack sugarcane, chrysanthemum, rice, cotton, soybeans, peanuts, and corn crops.

### 3.2. Deep Learning (DL) Algorithms

Deep Neural Networks (DNNs) are extensions of Artificial Neural Networks (ANNs) in terms of complexity, number of connections, and hidden layers. A CNN is a DNN that assigns learnable weights and biases to various aspects and objects of input images to distinguish and classify objects such as weeds. CNNs do not require manual feature selection; rather, the network learns important features automatically from training data to reveal useful information hidden. CNNs are robust at classifying various objects with different scales, orientations, and levels of occlusion. CNNs capture the spatial and temporal dependencies of the input image through relevant filters autonomously and hence provide better and more efficient image processing with a considerably lesser number of estimable parameters and processing time.

Max pooling is generally preferred as it discards the noise (data values unreliable for machine learning) in the data and performs the de-noising operation. Due to the possibility of saturation in sigmoid and tanh activation functions, CNNs employ the rectified linear activation unit (ReLU) as the activation function g(z), which outputs the aggregated value *z* if it is greater than 0, and 0 otherwise, i.e., g(z)=max{0,z}. Hence, it is a piece-wise linear function. The linearity of ReLU for z>0 allows it to preserve many properties of the input to facilitate stochastic gradient descent [15] and generalize to unseen data. Moreover, batch normalization is performed in CNN, which breaks up the training image data into mini-batches and standardizes the mini-batch input data to each layer. This stabilizes the learning process and considerably reduces the training epochs required to train CNNs [58].

One application of convolution and max pooling with ReLU forms one layer of the CNN pipeline. Typically multiple such layers can be employed. In each layer, we can have parallel processing based on different color channels or feature maps, for example, RGB. The output from the last pooling layer is flattened as a 1-D vector and fed to the fully connected layer (i.e., conventional Multilayer Perceptrons (MLP)) for image classification, e.g., detecting weeds within a given image. MLP outputs the probability of occurrence of each possible object (on which the CNN has been trained) through the softmax activation function $\sigma {\left(\vec{z}\right)}_{i}=e^{z_{i}}/\sum_{j=1}^{K}e^{z_{j}}$ where σ is the softmax function for *i*-th activation input vector ${\vec{z}}_{i}$, *K* represents the number of classes, and $e^{z_{i}}$ and $e^{z_{j}}$ represent the standard exponential functions for input and output vectors, respectively.

### 3.3. Variants of CNNs

The first usable and concrete CNN architecture was LeNet-5, proposed by Yann LeCun in 1998 and developed to recognize handwritten and printed characters [59]. It has a 2-layered architecture with 6 feature maps in the first layer and 16 feature maps in the second layer, followed by two fully-connected layers. A key outcome of this work is that larger image sizes can distinguish more pixels for the stroke end-points for written characters. After LeNet-5, ImageNet [60] has motivated researchers to propose enhancements leading to significant reductions in top-5 error percentages, i.e., the proportion of miss-classified images appearing in the top-5 results sorted in decreasing order of predictive confidence $P(Y_{i}|X_{i})$ (where $X_{i}$ is the input test data and $Y_{i}$ is the class label under consideration). For instance, AlexNet [61] is trained on 1.2 million images to achieve the lowest top-5 error rate of 16%, with five convolutional layers, followed by three fully connected dense layers. The authors used ReLU activation in all layers except the last layer, which employed softmax activation. Moreover, VGGNet 16 [62] is a deeper network than AlexNet with a top-5 error rate of 7.5%, with five CNN layers followed by three fully connected layers. VGGNet needs to estimate approximately 140 million parameters for training; however, due to the availability of pre-trained models, VGGNets are still being employed for several image classification tasks.

GoogleNet [63] achieved a top-5 error rate of 6.7% (almost equal to human-level performance) with its *inception modules* merging convolutional operations together rather than implementing them in different layers, and the concatenated output shows results from all convolutional operations. It employs 22 layers containing 9 inception module layers inserted between several pooling, convolutional, and fully connected layers with a drop-out layer used to drop input neurons from processing randomly to prevent over-fitting. GoogleNet achieves a significant reduction in the number of parameters to be estimated (4 million) as compared to AlexNet (60 million) and LeNet-5 (more than 100 million). Inception Module V1 used by GoogleNet was later upgraded to Inception Module V4 and Inception ResNet.

The ground-breaking research in CNN was achieved by ResNet (Residual Network) with a top-5 error rate of 3.6% (better than human performance) and remains unbeatable to date [64]. ResNet is a deep CNN with 152 layers which provides a solution to the vanishing gradient problem, i.e., the gradient becomes very small as it keeps on getting multiplied during backpropagation until it stops influencing any weight updates (learning stops). ResNet assumes that deeper layers should not generate more training errors than the shallower ones. Hence, it employs skip-connections, which transfer the results of a few layers to deeper layers while skipping some layers in between, hence preventing deeper layers from producing higher training errors than shallow layers. The gradient flow through the shortcut connection to the earlier layers, thus, reducing the vanishing gradient problem.

Along with this, SegNet [65] has been used for weed detection through image segmentation. It comprises an encoder network and a decoder network, much similar to Autoencoders [12]. In encoding, convolutions are performed using the 13 convolutional layers from VGGNet, followed by 2 × 2 max pooling to generate an encoded representation. In decoding, the max pooling indices from the encoding phase are employed to upsample the encoded data; for example, the 2 × 2 matrix is upsampled to a 4 × 4 matrix, and convolutions are also applied during the upsampling operations. Finally, the softmax function is applied at the end. In essence, there are no fully connected layers after decoding. Rather 1 × 1 convolutions are used, which allows outputting of a label for each pixel (a requirement for image segmentation) rather than a label for the whole input image. Such a setup is also called a Fully Convolutional Network (FCN). Another FCN-based algorithm is U-Net [66], which is used for biomedical image segmentation. It does not employ pooling indices during the decoding phase. Rather, the entire feature maps are transferred from encoder to decoder to acquire better segmentation performance but at the cost of time and memory. This makes U-Nets computationally intensive as compared to SegNets.

Finally, Deeplab [67] is a series of image segmentation algorithms invented by Google (Deeplabv1, Deeplabv2, Deeplabv3, Deeplabv3+) in 2018. The iterative application of pooling operations in FCNs reduces the spatial resolution of images. Deeplab uses atrous convolutions to generate much denser decoded feature maps with lesser computational overhead. It also enhances the localization accuracy in FCNs through the use of conditional random fields.

### 3.4. Machine Learning Algorithms

We now briefly describe the more important Machine Learning (ML) algorithms (for more details, refer to [68,69,70,71]). Support Vector Machines (SVMs) estimate an optimal hyper-plane between data points to linearly separate two classes by maximizing the margin with respect to the closest points called support vectors. Mathematically, from the equation y=m∗x+c, we can have y=a∗x+b and a∗x+b−y=0. Suppose we have vectors X=(x,y) and W=(a,−1), then the vector in hyper-plane become W∗x+b=0. Assume *n* training instances with each instance *x* of *D* dimension and belonging to class y=+1 or y=−1. Then the training would be $x_{i},y_{i}$ where i=1⋯n, $y_{i}\epsilon -1,1$, and $x\epsilon R^{D}$. If D=2, then hyper plane would be described as follows: for $y_{i}=1$ as $y_{i}\left(W*x+b\right)>=1$ and for $y_{i}=-1$ as $y_{i}\left(W*x+b\right)<=-1$. This leads to equations h1:w.x+b=−1 and h2:w.x+b=1 for two lines forming the hyper-plane. The distance between h1 and the starting point is (−1−b)/|w| and the distance between h2 and starting point is (1−b)/|W|. The maximum distance between h1 and h2 is called the margin *M*: M=(−1−b)/|w|−(1−b)/|w| = 2/|w|.

Decision Trees (DTs) model data as a tree whose nodes represent features as decision points, branches as feature values, and leaf nodes as class labels. Different patterns of label classification can then be extracted from the root node to each leaf node. At each decision node, features are selected at each node based on statistical criteria, mostly information gain. Specifically, the entropy of any partition of a dataset *D* can be expressed as $Entropy\left(D\right)=-\sum_{i}^{n}p_{i}*log_{2}\left(p_{i}\right)$, where *p* is the probability of occurrence of an instance *i* in *n* total instances. The Information Gain G(D,A) represents the change in entropy of *D* when we consider feature *A* for decision node: $G\left(D,A\right)=E\left(D\right)-\sum_{i}^{f}\left(\right|D_{f}|/|D|)*E\left(D\right)$, where *f* represents all possible values of *F*, |D| represents total instances in D, and $|D_{f}|$ represents the number of rows containing the particular value *f*. Random Forest (RF) is a well-known DT ensemble algorithm that employs bootstrap aggregation (bagging) to generate *m* (m<N) datasets $D_{1},D_{2},\cdots ,D_{m}$ by sampling *D* uniformly and randomly with replacement. A set of *m* DTs $h_{1},h_{2},\cdots ,h_{m}$ is generated for each dataset. An unseen instance is then tested on each tree, and the class with the majority vote from amongst all *m* trees is output as the final predicted value.

In the same vein, Adaptive boosting (Adaboost) uses an ensemble boosting technique to construct a strong learner from a number of weak learners, which are typically DTs. In each iteration, it adapts by finding miss-classified data points from each learner and increases their weights (to learn them with more emphasis in the next iteration) while decreasing the weights of correctly classified points (to learn them with less emphasis in the next iteration). As long as the performance of each learner is better than random guessing, Adaboost is guaranteed to converge to a strong learner. A boosted classifier over *T* weak classifiers can be represented as $F_{T}\left(x\right)=\sum_{t=1}^{T}f_{t}\left(x\right)$ where each $f_{t}$ is a weak learner that takes *x* as input and outputs the class label. A hypothesis $h(x_{i})$ for each sample in training data is output by each $f_{t}$. At iteration *t*, each weak learner is assigned a coefficient $\alpha_{t}$ to minimize the following sum of training error $E_{t}$: $E_{t}=\sum_{i}E\left[F_{t-1}\left(x_{i}\right)+\alpha_{t}h\left(x_{i}\right)\right]$ where $F_{t-1}$ is the boosted classifier of the previous stage, E(F) represents the error function and $f_{t}\left(x\right)=\alpha h\left(x\right)$ is the weak learner under consideration for addition to the Adaboost classifier.

Artificial Neural (ANNs), more commonly known as Multilayer Perceptrons, are ML versions of the CNNs described earlier. They can model complex non-linear stochastic relationships between predictors and label through layers of neurons (processing units). Predictor data are fed to an input layer, processed over one or more hidden layers, and predictions are generated at the output layer. The neurons between each pair of layers are connected to each other through synapses called *weights*. The weight vectors are updated based on numerical values output from a mathematical *activation function* from each neuron in the hidden and output layer, based on the aggregated input at each neuron. A sample output can be represented as $h_{i}=\sigma \left(\sum_{j=0}^{N}V_{ij}x_{i}+T_{i}^{h}id\right)$ where σ is the activation function, *N* is the number of input neurons to a given neuron, $v_{i}j$ are the weights of these input neurons, $x_{i}$ shows the input values to input neurons, and *T* is the threshold for activation.

Moreover, the k-nearest neighbor (KNN) categorizes the input *x* by its *k* nearest neighbors. For *k*, it will observe the adjacent neighbors of hidden data points and assign the data point to a class with the highest number of data points from all classes of *k* neighbors. It uses Euclidean distance when it calculates the probabilities. KNN gives the input *x* to the class which has the highest probability: $P\left(Y=j|X=x\right)=1/K\sum_{i\epsilon a}I\left(y_{i}=j\right)$, where *a* is the set of *k* nearest neighbors and $I(y_{i}=j)$ is an indication variable which calculates to 1 if a given neighbor $\left(x_{i},y_{i}\right)$ in *a* is a participant of class *j*, else it calculates to 0. Finally, K-means is primarily used for cluster analysis. It divides the data into *k* predefined unique clusters (collection of data points with similar features) where each data point should preferably belong to only a single cluster. It initially sets *k* centroids randomly and assigns every data point to its nearby cluster. It calculates the centroids for each cluster by averaging all the data points belonging to that cluster. The Euclidean distance between a data point *q* and centroid *p* is typically calculated as $d\left(q,p\right)=\sqrt{\sum_{i=1}^{n}{\left(q_{i}-p_{i}\right)}^{2}}$, where *n* is the the number of features.

### 3.5. Image Processing Techniques

Image Processing (IP) can be used to improve images for further processing with DL or ML algorithms. IP facilitates algorithm tasks by improving image quality and transforming images to meet the needs of the algorithm. Local Binary Patterns (LBP) [72] is a visual descriptor of images in CV based on thresholding. It divides an image into equal-sized cells, for example, with each cell containing 16 × 16 pixels. Each *center* pixel *c* is then compared to each of its 8 neighbors *n*, for example, clockwise starting from top-right, middle-right, then bottom-right, and so on. The thresholding works as follows: if the value of *n* is greater than the value of *c*, we set n=0; otherwise, we set n=1, giving us an 8-digit binary number. Then, we compute a histogram over *c* indicating the frequency of each of the 256 ($2^{8}$) combinations of this binary number. Finally, we concatenate the histograms of all cells to form a feature vector for the whole image, which can then be processed in machine learning and deep learning tasks. Mathematically, LBP for pixel *c* over a neighboring radius *r* (set to 8) is estimated as follows: $LBP_{c,r}=\sum_{i}^{P-1}s\left(v_{i}-v_{c}\right)2^{i}$ where *P* is the number of neighboring pixels, $v_{i}$ and $v_{c}$ are values of the neighboring and center pixel respectively, s(t) is thresholding such that s(t)=1 if t>0 and 0 otherwise. The histogram feature vector of size $2^{P}$ (256) is then estimated from the obtained LBP code.

Moreover, Simple Linear Iterative Clustering (SLIC) [73] is an image segmentation algorithm that uses k-means clustering to create *superpixels*, which are small-sized clusters of pixels sharing common features. Clustering is done by distance measurement computation in 5D (labxy) space, where (*l*, *a*, *b*) is the 3-dimensional color representation of the pixel at coordinate (x,y). The distance measure $D_{S}$ is then defined as follows: $D_{S}=d_{lab}+\left(m/s\right)*d_{xy}$ where $d_{lab}=\sqrt{{\left(l_{k}-l_{i}\right)}^{2}+{\left(a_{k}-a_{i}\right)}^{2}+{\left(b_{k}-b_{i}\right)}^{2}}$ and $d_{xy}=\sqrt{{\left(x_{k}-x_{i}\right)}^{2}+{\left(y_{k}-y_{i}\right)}^{2}}$ and *m* controls the density of superpixels proportionally. Moreover, Histogram of Gradients (HoG) [74] is used for feature extraction. It divides an image into a number of *regions*, and for each region, it estimates the gradient (magnitude) and the orientation (direction) of the edges in that region. Then, the histogram of this data (HoG) for each region is generated separately. Suppose that for each pixel (x,y), we define HG as the distance between the adjacent right and adjacent left pixel values and VG as the distance between the adjacent top and bottom pixel values. The gradient magnitude GM is $GM=\sqrt{{\left(HG\right)}^{2}+{\left(VG\right)}^{2}}$ and the gradient angle GA is $GA=tan^{-1}\left(HG/VG\right)$ Then, HoG is generated by binning the frequencies of either GM or GA or both together [74].

In addition, Hilbert Transform (HT) [18] is used to separate features of a specific shape within an image, for example, circles, lines, and ellipses. A line is a collection of single points with slope *m* and intercept *c* and y=mx+c in the xy plane. In HT, we convert a line from (x,y) plane to (m,c) space, i.e., from y=mx+c to c=−mx+y. To avoid unbounded values of *m*, the well-known Hough space (r,θ) transformation can also be used as follows: r=x.cosθ+y.sinθ. Moreover, Median filtering [17] is a non-linear IP technique that maintains edges while removing noise. It calculates the median gray-scale value of a pixel’s neighborhood. In applying a fixed-size kernel, we sort all pixel values within this kernel based on gray-scale values. Then, the median value of this sorted array will be used, and zeros can be padded in rows and columns to complete the pixel count. Finally, Background Subtraction (BS) [75] is a well-known technique used in IP and CV for detecting moving objects in videos from static cameras for additional processing. It isolates these foreground objects with respect to a reference image by subtracting the current frame from a reference frame called the background model. If the data points are non-linear, then we need to add one more dimension to the data point, which will be $z=x^{2}+y^{2}$.

GANs (generative adversarial networks) [76] are types of generative deep learning algorithms whose purpose is to learn a set of training samples and their probability distributions and then generate data from this distribution. GANs can produce more samples based on the measured probability distribution and are particularly accurate in producing realistic high-resolution images. GANs comprise two different feed-forward artificial neural networks named Generator (Gen) and Discriminator (Dis) that participate in an adversarial game. The input to the generator is Gaussian noise $p_{z}\left(x\right)$ and Gen tries to generate an approximation $p_{model}\left(x\right)$ to the probability distribution of the actual data $p_{data}\left(x\right)$. Meanwhile, Dis learns to distinguish whether a data point *x* is sampled from $p_{data}\left(x\right)$ or $p_{z}\left(x\right)$, the latter being input to Dis as data sampled from $p_{model}\left(x\right)$. The task of Gen is to fool Dis into thinking that data sampled from $p_{model}\left(x\right)$ is actually the data sampled from $p_{data}\left(x\right)$. Therefore, Dis maximizes the probability of classifying data as $p_{data}\left(x\right)$ and minimizes the probability of classifying it as $p_{model}\left(x\right)$, while Gen tries to do the exact opposite. In this context, both Dis and Gen participate in a two-player minimax game with the value function Val(Dis,Gen) as $min_{Gen}max_{Dis}Val\left(Dis,Gen\right)=E_{x}∼p_{data}\left(x\right)\left[logDis\left(x\right)\right]+E_{z}∼p_{z}\left(z\right)\left[log\left(1-Dis\left(Gen\left(z\right)\right)\right)\right)]$ where Gen maximizes logDis(Gen(z)) rather than minimizing log(1−Dis(Gen(z))) [12,76].

## 4. SLR Methodology

In this section, we address RQ1: What is the trend of employing deep learning to address the problem of weed detection in recent years. To answer this, we conduct an SLR by following the standard methodology [77] and dividing our work into three phases: (1) Planning, (2) Execution, and (3) Reporting (see Figure 8).

In the planning phase, we identify the research objective, research protocols, search keywords, and digital sources for extracting the relevant papers. In the execution phase, we execute our search queries on each of the identified digital sources to acquire the relevant corpus of papers by using a three-step technique (described below) and eliminating duplicates. In the reporting phase, we apply thematic classification to our final list of extracted papers and describe them in detail, identify the limitations of these works, and then propose a concrete set of future work recommendations.

In this SLR, our specific consideration is in the domain of DL applications in weed detection. Our research objective is to extract the state-of-the-art, identify published academic research related to this domain, understand the content of these papers, classify our results using different methods of analyses, identify the gaps or limitations through these classifications, and consequently propose guidelines and directions to motivate and enhance the state-of-the-art research. To achieve this, we adopted the following inclusion and exclusion criteria.

We targeted original academic research content published in journals, conferences, workshops, and symposiums. We excluded periodicals (magazines and news from newspapers), letters, books, and online content, specifically websites, blogs, and social network feeds.We considered papers published in the English language only.We selected the following digital sources: IEEE, ACM, Elsevier, Springer, and Google Scholar. Our previous experience [78,79] has shown us that these sources are collectively effective in retrieving required content for data analytics, machine learning, and any computer science domain in general. Moreover, Google Scholar can effectively index published data from other sources, for instance, Taylor and Francis, Wiley, MDPI, and Inderscience.Initially, we focused on research published from 2010 and onwards. However, after some preliminary results, we discovered that the content most relevant to the domain of weed detection using DL was published primarily from 2015 onwards. Therefore, we focus our SLR from January 2015 till January 2023.

The published DL research has seen an exponential rise in different domains in the last several years, for example, after the proposal of generative adversarial networks in 2014 and the discovery of different variants of CNNs, autoencoders, and recurrent neural networks (RNNs). Hence, we decided to focus only on the latest research from 2015 till January 2023. We considered all articles irrespective of the country of the first authors. Moreover, we considered three types of research publications: (1) application papers, i.e., papers that present a novel research idea along with experimental results; (2) literature reviews (both systematic and non-systematic) and (3) frameworks, i.e., papers that present a novel research framework/idea with a concrete design but it has not been validated with experimental work.

We formulated our search queries from our four research questions (mentioned in Section 1). In these questions, we focused on discovering important information on smart farming, particularly deep learning, applications for weed detection, for instance, the different research trends and statistics, types of weeds detected and algorithms used, and performance comparison of algorithms. From our previous experience [78,79], we concluded that all this information could be extracted by using search queries based on different combinations of the following three keywords: (1) weed detection, (2) smart farming, which is used interchangeably with *smart agriculture* and *precision farming*, (3) Weed Classification, and (4) deep learning, in which we particularly targeted CNNs. Based on this, we initially executed the following nine search queries (& = AND): (1) {“weed detection”}, (2) {“precision farming” & “weed detection”}, (3) {“precision agriculture” & “weed detection”}, (4) {“smart farming” & “weed detection”}, (5) {“weed detection” & “deep learning”}, (6) {“precision farming” & “weed detection” & “deep learning”}, (7) {“precision agriculture” & “weed detection” & “deep learning”}, (8) {“smart farming” & “weed detection” & “deep learning”}, (9) {“weed detection” & “CNN”}, (10) {“weed classification”}, (11) {“precision farming” & “weed classification”}, (11) {“precision agriculture” & “weed classification”}, (12) {“smart farming” & “weed classification”}, (13) {“weed classification” & “deep learning”}, (14) {“precision farming” & “weed classifcation” & “deep learning”}, (15) {“weed classification” & “CNN”}.

The results from these queries showed that the most relevant papers could be obtained only through the following two queries (which we also used in our SLR): (1) {“weed detection” & “deep learning”} (labeled as Q1), (2) {“weed detection” & “CNN”} (labeled as Q2) and (3) {“weed classification” & “CNN”} (labeled as Q3). Q1 and Q2 also retrieved articles related to applications in ML and IP (without any DL implementation) for weed detection. We considered these papers in our SLR to facilitate a comparison of these algorithms with DL to understand further the strengths and limitations of these approaches (RQ4 in Section 1).

We implemented a three-step procedure to filter out our required subset of research articles (shown in the execution phase in Figure 8). In the first step, we filtered out the articles based on their titles, i.e., we did not consider articles whose titles were not related to the domain of weed detection using DL, for instance, several titles related to smart farming but no research contribution to weed detection. In the second step, we adopted the same approach to filter articles from the first step based on their abstracts, and in the third step, we filtered articles from the second step based on their content, i.e., after reading the article’s introduction, methodology, and results section.

Across all the digital sources, we filtered out a total of 129 articles from the title filtration step, of which 25 were duplicates. Thus, we filtered out 92, 81, and 64 articles after the title, abstract and content filtration, respectively. The breakdown of these numbers with respect to each digital source (IEEE, ACM, Elsevier, Springer, Google Scholar) and search query (Q1, Q2, and Q3) is shown in Table 1. Of our 64 articles, 49 (83%) were retrieved by Q1 alone, 10 by Q2, and the remaining 4 by Q3. Across Q1, Q2, and Q3, IEEE retrieved 29 (53%) of these 64 articles, while ACM retrieved 2 (3%) only. The 8 articles were retrieved from Springer, 10 from Elsevier, and 15 from Google Scholar. All the above trends are also applicable for title filtering and abstract filtering data.

Figure 9 shows the frequency distribution of our filtered 64 articles from January 2015 to January 2023. We observed an exponential trend in the number of publications from 2015 onwards. Moreover, Figure 10 demonstrates that out of our 64 articles, 55 were application papers, 8 were literature reviews, and only 1 article introduced a framework. Finally, Figure 11 shows the co-author citation graph for our 64 papers. Out of a total of 221 authors in these papers, the presented 13 authors in Figure 11 have the strongest co-authorship links. The colors red and green represent two clusters of co-authorship links with the author Arnold W. Schumann participating in the red cluster (in the years 2020 and 2021) [38] as well as in the green cluster (in 2019) [80,81].

## 5. Review of Weed Detection Algorithms

In this section, we summarize the findings of our surveys on application articles and literature reviews. For convenience, we merged the single paper, which proposed a framework [82] into the application articles (i.e., implementation) category.

### 5.1. Weed Datasets Available for Deep Learning

In this section, we answer RQ2: Which types of weeds and corresponding crops have been detected using deep learning, and what are the characteristics of the corresponding weed datasets? In this regard, we extracted and classified important characteristics regarding the datasets of weed images used by researchers of the application papers, shown in Table 2. These are (1) corresponds to the reference article, (2) the dataset name or label (Dataset), (3) the size or the number of images in the dataset (Size), (4) the type of crop for which the weed was detected by the authors (Crop), (5) the particular weed type which was detected (Weed), (6) the modality of the dataset (Modality), (7) the data collection technique through which images were acquired (Data Collection), and (8) the resolution of the images (Resolution). We use N/M (Not Mentioned) to indicate any of this information not mentioned by the authors.

Table 2 shows that researchers in application papers have used a total of 44 unique weed image datasets. Under `Dataset,’ we present the dataset names as labeled by the corresponding researchers in their papers. Where no specific name was assigned to a dataset by the authors, we have used the city’s name wherein the implementation was done as dataset name (e.g., Gharo (Pakistan)). Most researchers have created their personalized weed image datasets by using drones, robots, and a large variety of cameras. However, three available datasets have also been used: (1) Indian Agriculture Research Institute (IARI), (2) Crop Weed Field Image (CWFID), and (3) Plant Seedling (available on Kaggle). The three largest datasets are Gold Field (36,000), WeedMap (11,441), and Bok Choy (11,150), while the three smallest datasets are CWFID (120), Tobacco Field (76), and IARI (60).

A total of 26 unique crops are used in weed detection. Notably, sugar beet is the most common crop (used in 13 papers), followed by carrot and wheat (in 5 papers), maize (4 papers), and rice, soybean, corn, and cereal (in 4 papers). All the remaining crops are used once in our filtered 51 application papers. As far as the distribution of 34 weed types is concerned, Dockleaf is used for weed detection most frequently (in 7 papers), followed by fat-hen (5 papers), Canadian thistle, Chickweed, and Mayweed (in 3 papers), and Blackgrass, Jungle rice, Nutsedge, Shepherd’s purse, Fescue grass, Grass weed and Pigweed (in 2 papers). The remaining 22 weeds are used once. Regarding the input modality, RGB is used most frequently in 33 articles, 5 times in combination with Near Infrared (NIR), and 5 times in multi-spectral mode. Moreover, hyperspectral, NIR, grayscale, and NDVI modes are used once, while Color Infra Red (CIR) is employed twice. Regarding data collection, drones (as UAVs) are used to acquire images in 13 papers, cameras are used in 22 papers (with different types such as digital, cellphone, and professional), robots are used in 9 papers, and the remaining are referred to as datasets. Here, we would like to mention about Bonirob, an agricultural robot developed in Germany and used to acquire images in 3 articles. A variety of resolutions are also employed, ranging from a minimum of 61 × 61 and 64 × 64 to a maximum of 4512 × 3000 and 4000 × 3000. Out of 51 papers, 14 have used a resolution greater than 1000, and the remaining 36 have a resolution less than 1000. Most high-resolution images have RGB Modality, and low resolution have NIR, CIR, Greyscale, hyperspectral, and NIR + RGB. High-resolution images are mostly snapped by drones, while low-resolution images are snapped through semi-professional or cell phone cameras. Only 35 articles have mentioned the size of the dataset, with the GoldField dataset having the most images (36,000) and CWFID being used in two papers with the fewest images (60).

### 5.2. Algorithms Used for Weed Detection

In this and the next section, we collectively answer RQ3: which deep learning algorithms are best suited for a particular weed/crop combination? In this regard, we initially extract and classify the algorithmic performance of our application papers, shown in Table 3. Here, we present (1) the article reference (Article), (2) the name of the algorithm(s) employed by the authors (Algorithm), (3) the image processing technique used, if any (IPT), (4) performance measure or KPI, such as accuracy or precision/recall (KPI), (5) the maximum value of this KPI achieved by the authors (Result), (6) the training time taken by the algorithm (TR.Time), (7) the split ratio for train and test sets (TR.TS.Split), and (8) a thematic classification label (TCL) which we assigned to each paper based on the algorithm and the weed detection approach used in the paper.

We found that both ML and DL algorithms are used by researchers for weed detection. In total, 10 different algorithms are used whose frequency distribution is shown in Table 4. For DL, CNN is used in general. Although Autoencoder (AE) is also used, as such, no specific AE has been used separately, for example, denoising AE, stacked AE, or variational AE. Rather, only the AE architecture is used within the algorithmic process of CNNs to improve the CNN performance. Hence, we have not considered AE as separate from CNN in this work. The CNN algorithms are applied 65 (with different variants) times in application papers (details below), followed by ML algorithms, specifically, SVM (12 times), RF (5), ANN (5), KNN (2), Boosting (XGBoost and Adaboost) (2), DT (includes vanilla DT and ERT) (2), NBG (1), and LR (1). This shows that many representative ML algorithms have already been applied for weed detection, although the frequency of applications remains severely limited as compared to CNN.

The 65 CNNs appeared over a total of 19 different variants, whose frequency distribution is shown in Table 5. Specifically, the custom CNN model is used in 12 implementations, and transfer learning is used in 21 implementation articles, including 7 implementations of ResNet, SegNet, and VGGNet. GoogleNet and FCN are used in 6 and 5 articles, respectively. YOLO, U-Net, and AlexNet are each used in three articles while DetectNet, DeepLab (all variants combined), and SSD are used in two articles, and Mask R CNN, CaffeNet, CenterNet, GCN, MobileNet, DenseNet, and Xception are used once. This demonstrates that researchers are not focusing on several CNN models only; rather, there is a trend to explore recently-introduced CNN variants as soon as they are published (as most of the variants are introduced recently and have very limited applications).

Image Processing (IP) techniques are used in 38 application articles, with a total of 16 unique techniques whose frequency distribution is shown in Table 6. There are four (4) articles in which the authors applied only IP techniques for weed detection, i.e., without using any ML or DL algorithm. In these articles, the authors have used techniques such as both background and foreground subtraction (BFS), converting RGB to Grayscale, binarizing and labeling the images (CV), feature extraction (Ftr Ext), classification (Classify), and image tiling techniques. In the remaining 34 articles, IP techniques have been used for pre-processing images for either a separate application of ML or DL or both ML and DL collectively. Table 6 shows that feature extraction, image segmentation, and BFS are used more frequently, along with SLIC, LBP, and HoG, while some less-applied techniques are also used once, for example, morphological erosion and histogram equalization (HE).

Regarding the use of performance measures, 32 out of the 51 application papers used the accuracy of weed detection, which gives an average of the predictive performance for both classes (`weed detected’ and `no weed detected’ respectively). Precision and recall for the `weed detected’ class, which provide a better indicator of the weed detection performance, are employed in only 11 articles. Of the latter, two papers employ the F1-Score, which is estimated from both precision and recall values. Performance measure was not m1entioned in four articles, while Area Under the Curve (AUC), which provides a measure of overall classification performance similar to accuracy, is used in two (2) articles, once in combination with F1-score. In total, 11 papers out of 52 mentioned model training timing, YOLO with K-means took the shortest time to train at 6.8 ms, and CNN took the longest at 1895 min. As there is no standard for dividing data into train and test, most papers used 60–40, 70–30, and 80–20 splits.

Here we addressed How does the performance and usage of machine learning and image processing techniques compare with that of deep learning for weed detection? where we categorize each application paper by assigning it a unique thematic classification label (TCL) based on the usage of ML, DL, and IP, along with other information such as training time and KPI. We classify application paper under five TCLs: (1) **DL.AE**, (2) **ML**, (3) **DL.CNN**, (4) **IP**, and (5) **ML.DL.CNN**. In the following, we define these TCLs and present data about their constituent articles. A summary of this is presented in Figure 12. (the details of the articles are provided in Section 6).

**DL.AE:** This type represents five articles that employ the autoencoder (AE) technology, i.e., the encoder-decoder DNNs for weed detection. In two of these articles, both the encoder and decoder are modeled as an FCN. In the remaining three articles, two articles use the AE-based CNN called SegNet as {SegNet256, SegNet512} and {SegNet, U-Net, DeepLabV3}, while the last article uses only DeepLab. We remark that the U-Net and DeepLab series of CNN variants are also based on AE. Histogram Equalization (HE) is the only technology that is specified for DL.AE category.**ML:** This type represents 9 articles that only employ ML techniques for weed detection. In four of these articles, the performance of a set of ML algorithms are compared, while an individual ML algorithm is used in the remaining five articles. The comparison sets include {SVM, ANN} (in two articles), {NBG, DT, KNN, SVM, ANN} (1) and {ERT, RF} (1). The individual applications include RF (2) and SVM (3). In one of the SVM applications, K-means clustering is used to pre-process the image data and then IP techniques such as SLIC, BFS, Masking, Feature Extraction, Markov Random Field, Image Segmentation, and Morphological Erosion. are used with ML**DL.CNN:** This type represents the 31 articles that employ the CNN or one/more of its variants for weed detection. Regarding individual applications, CNN is used most frequently in nine articles, GoogleNet in two articles, and YOLO, SSD, AlexNet, and CenterNet in two articles each. Besides this, the following combinations of variants are used: {ResNet, U-Net, SegNet, FCN, GoogleNet}, {GCN, AlexNet, VGGNet, ResNet}, {SSD, VGGNet}, {CNN, VGGNet}, {SegNet, VGGNet, U-Net}, {DetectNet, GoogLeNet VGGNet}, {ResNet-50, YOLO}, {SegNet, SegNet-Basic}, {Mask R CNN, FCN} and {YOLO, GoogleNet}. All these combinations are used once, except {DetectNet, GoogLeNet VGGNet}, which is used twice. The IP techniques employed with DL.CNN papers are as follows: HoG, LBP, BFS (background), image labeling, image segmentation, CV, feature extraction, and clustering.**IP:** This type represents four articles that only employ IP techniques for weed detection. As mentioned above, these techniques are image tiling, classification, CV, feature extraction, and background and foreground subtraction.**ML.DL.CNN:** This type represents seven articles that employ both CNN (or one/more of its variants) and ML for weed detection. These 7 combinations are as follows: {SVM, ANN, CNN}, {ResNet, SVM}, {SVM, KNN, ESD, CNN}, {CaffeNet, SVM, Adaboost, RF}, {ResNet, SVM, RF}, {AlexNet, ANN} and {ResNet, VGGNet, MobileNet, DenseNet, Xception, SVM, XGBoost, LR} (abbreviations of algorithms are shown for this sequence for brevity). The IP techniques used in ML.DL.CNN includes MGF, HT, SLIC, and CV algorithms.

In this section, we analyze the performance of algorithms with respect to our TLCs. We will first emphasize the training time and train-test split ratio data shown in Table 3. Our purpose in listing the training time was to gauge the delay or speed-up achieved in training the DL algorithms due to the complicated nature of this learning task. In fact, GPU usage has addressed this problem thoroughly. It is considered prudent to conduct a comparative analysis for different tasks and GPU/CPU settings. In our case, no training time was recorded by the authors in 36 (out of 51) articles. In the remaining articles, the maximum recorded time is as follows: (1) 480 s (IP), (2) 83 s (ML), (3) 976 m (DL.CNN), (4) 480 s (ML.DL.CNN), and (5) 11,389 s (DL.AE). Thus, the maximum training time is 976 min (approximately 16 h) for DL.CNN followed by 3.1 h for DL.AE. Where pre-trained DL models were employed, training time is in the order of seconds.

The train-test split ratio has been mentioned in 32 articles. After analyzing the ratios, we mapped them into four categories: 60–40, 70–30, 80–20, and 90–10. Out of 31 articles, 11 articles used an 80–20, and 11 others used a 70–30 ratio (these two ratios are the most common ones in ML and DL domains). However, four articles based on DL have used a 90–10 ratio, and five articles (three from ML and two from DL.CNN) have used a 60–40 ratio. Our aim here is to present such information. However, we are not intended to make conclusions about the reason for a ratio selection because these matters are guided by several factors, such as the size of the dataset, previous experience, or trial-and-error experimentation.

To compare the performance of algorithms across our TCLs, we executed the following steps. We decided to analyze all the different performance measures together by focusing on their values. In other words, we do not specifically distinguish between accuracy, precision, recall, etc., but we focus only on their values to compare all application papers together. For this, we analyzed the values in Table 3 and manually created three bins or ranges of values: (1) [45–85) (labeled as low performance), (2) [85–95) (medium), and (3) [95–100] (high). Moreover, we estimated and considered the F1-score in case both precision and recall are measured, and we considered the performance values of each algorithm separately in case multiple algorithms were used in the article. Finally, we considered the F1-score over AUC in articles where both were recorded, and for a recorded performance of `>90’ by the authors, we considered it in the range of 85–95. The results are shown in Figure 13 and discussed as follows:**ML:** 2 algorithms have a low performance, 4 have medium, while 10 have a high performance (over multiple algorithms).**IP:** Two articles mention their results which are low and high, respectively (no multiple algorithms used).**DL.CNN:** 5 algorithms have a low performance, 21 have a medium, and 18 have a high performance (over multiple algorithms).**ML.DL.CNN**: 2 algorithms have a low performance, 13 have medium while 10 have a high performance (over multiple algorithms), and**DL.AE:** One article has a low performance, two have a medium, and the remaining two demonstrate high performance.

A notable trend here is that, although ML applications are limited with respect to DL.CNN, the proportion of algorithms that demonstrate a high performance is more (63%) as compared to DL.CNN (44%). In ML.DL.CNN category, of the 10 algorithms demonstrating high performance, 4 of these are ML algorithms while the rest are CNN variants. This clearly shows that traditional ML algorithms, particularly SVM and RF, are demonstrating performances at par with CNN and its variants. Regarding the use of the AE, the performance seems to have remained consistent over the low, medium, and high bands. However, we do acknowledge that a dataset of five articles is limited to making any generalization of this trend.

## 6. Classification as per Weeds/Crops

To understand the aforementioned trends more thoroughly, we created a table shown in Table 7, which illustrates associations of different weed types and different crop types respectively with their algorithmic applications and demonstrates which algorithm has been used to detect the weed type which belongs to a crop type.

We would like to highlight the following facts before our analysis: (1) We do not present the algorithms for which their corresponding weed/crop types are not mentioned by the authors; (2) In all papers using multiple weed types, the performance achieved is applicable for each type; (3) If algorithm A is used as a feature extractor for algorithm B (e.g., {ResNet-50, YOLO}, {RCNN, SSD} and {VGGNet, SSD})), we have assigned A and B the same performance measure as both contributed to achieving this performance, (4) We do not specifically distinguish between accuracy, precision, recall, etc. as mentioned above for Figure 13, and (5) we show the highest accuracy with which the weed belonging to a crop type is detected using one or more algorithms and in each algorithm. The table assists in understanding the applications and deriving appropriate recommendations.

The performance of detecting each of the weed types in different crops is high, with a minimum performance of 84% for Field Pansy and Wheat crops and 99% for Bluegrass in Corn, Cockleblur, Dockleaf in Soybean, Pasture, Turfgrass, and bahiagrass, Japanese Hop in Maize, Wheat and Peanut and Black Nightshade and Velveleaf in Tomato and Cotton. The top-4 weeds that are used most frequently by researchers are Canadian thistle, followed by Dockleaf, Fat-hen, and Velvetleaf, while those who are applied only once include Latana Camara, Chinee Apple, SnakeWeed, Indian Jointvetch, Hedge Bind, Benghal Dayflower, Grass Weed, Dicot, Paragrass, Crowfoot and CockleBlur.The top-3 crops that are used most frequently by researchers are Sugar beet, followed by Carrot and Maize, while those that are applied only once include Cranesbill, Charlock, Soybean, Canola, Peanut, White Cabbage, Tobacco Seedling, Lettuce, Tomato, Radish. It is apparent that the applications of the top-4 weed types are spread out over different algorithms, i.e., the novelty in research articles is introduced through the application of novel or other algorithms. From the perspective of algorithms, each algorithm demonstrated high performance in detecting one or more weed types, with a minimum performance of 87% for FCN and SSD and a maximum of 99% for SVM, CNN, VGGNet, and DetectNet. For DL, the top-3 algorithms that are used most frequently are CNN, VGGNet, and FCN, while for ML, the top-3 are SVM, ANN, and KNN. The average performance over all DL algorithms is 94% while the same performance over all the ML algorithms is 90%. In our opinion, this difference is small, and we believe the ML community still has much to offer for weed detection, which could be potentially comparable to DL algorithms. Furthermore, when combined with other algorithms, SVM and Mask R CNN performed equally or better (five times) and only underperformed once (to YOLO). SVM and YOLO have not been used together in other cases. YOLO is the only algorithm that performs every time better or equal to other algorithms when combined. In three cases, CNN alone performed best, and when combined with other algorithms, it performed equally (three times). The following are the findings regarding crop association with algorithms. Sugar beet was used as a crop in 13 of the 51 papers, and different weed types were associated with each paper. Papers with sugar beet crops used a combination of FCN and Mask R CNN six times and FCN, Mask R CNN, and CNN four times. YOLO was only used once to detect sugar beet, and it had the lowest accuracy of 89 percent when compared to FCN, Mask R CNN, and CNN. In nine papers, sugar beet is mostly combined with maize and wheat. Maize and rice were used together four times, and GoogleNet and SSD were frequently used to detect weeds in maize and rice. GoogleNet and SSD are used together twice, and SSD and GoogleNet are used separately once. GoogleNet provided the highest accuracy of 99 percent when combined with SSD and without, but SSD alone provided 98 percent accuracy. Tomato and cotton were used together in two cases where ResNet, VGGNet, MobileNet, DenseNet, Xception, and SVM were used to detect weeds, and both times SVM and DenseNet provided 99 percent accuracy. Furthermore, each crop is used in a unique combination.

## 7. Summary of Identified Articles in SLR

This section describes a literature survey of our identified 60 articles. We present the application papers and literature review papers in separate sections.

### 7.1. Summary of Identified Application Papers

In this section, we briefly discuss each application paper according to our TCLs. The sequence in which these papers are discussed is the same as the one found in Table 3.

#### 7.1.1. IP Papers

In [32], the basic idea adopted by the authors is to detect weeds at different stages of growth of the wheat crop, along with detecting the barren land to determine the amount of land used for cultivation. For detection, the authors employ background subtraction techniques in the Hue Saturation Value (HSV) color space, but they can only achieve a maximum weed detection accuracy of 67% with high-resolution images acquired through drones. Moreover, in [84], the authors use CV functions for the classification of weeds and crops, notably, *rgb2gray* for detection of green plants, *im2bw* to convert digital images to binary images, *bwlabel* for labeling binary images, and *regionprops* for measuring feature of images and detection of weed. The classification accuracy obtained with these functions is 99% with a training time of approximately 3 s.

In [82], authors create and implement a framework called the Image Processing Operation (IPO) library for the classification of weeds. IPO stores information about weeds and crops in JSON format which are then automatically converted to MATLAB functions to perform weed discrimination, with the option to add personalized user-defined functions. The authors claim that IPO is partially successful and discuss methods to remove some of its limitations. Finally, in [95], the authors study different features of weed leaves for detection using IP. In their method, the authors propose the execution of several stages in sequences, such as *foreground extraction* with grey-scale images, image tiling, feature extraction, and classification. For classification, authors employ moment-invariant shape features i.e., rotation, scaling, and translation for identifying the weed, with a training time of 480 s.

In [131], the researchers described the importance of large datasets for better weed detection and also emphasized the need for GANs. They also mention a lack of real-world datasets for weeds. To solve this problem, they proposed a model that combines transfer learning or a pre-trained model with GANs. The crop weed dataset at the early growth stage was used, with 202 images of tomato as a crop and 130 images of black nightshade as a weed. To select the best parameters for the model, various combinations of hyperparameter tuning were used. Three pre-trained models were used: Xception, Inception-ResNet, and DenseNet. Xception outperformed with a 99.07% accuracy.

Researchers in [132] study combined Generative Adversarial Networks (GANs) with Deep Convolutional Networks to create a model that detects weed better than existing models. GANs are used to generate synthetic images of weeds, and deep neural networks are used to detect weed images from original and GAN-generated images. They also compared their model to existing models like AlexNet, ResNet, VGG16, and GoogleNet, but their model outperformed with an accuracy of 96.34%.

Researchers focused on a robust image segmentation method in [133], which will be used to distinguish between crops and weeds in real time. They also discussed using annotated images in various studies and stated that annotating images could be time-consuming. However, they used GANs to generate synthetic images to supplement the dataset. Then, for image segmentation, they used CNN variants such as UNet-ResNet, SegNet, and BonNet. UNet-ResNet and SegNet outperformed with 98.3 percent accuracy.

The authors of [134] study developed an algorithm that is used to synthesize real agricultural images. The images were captured with a multi-spectral camera, and Near-Infrared images were collected. They used conditional GAN for segmentation. They also stated that their experiments improved the generalization ability of segmentation and enhanced the model’s performance. They used various CNN variants for segmentation, including UNet, SegNet, ResNet, and UNet-ResNet, and UNet outperformed with 97% accuracy in Crop detection and 72% in Weeds.

#### 7.1.2. ML Papers

In [90], the authors attempt to classify soil, soybean, and weed images based on the color indices of these three classes. They compare the performance of SVM and ANN over this task after processing and segregating the datasets through SLIC. The results do not demonstrate any major difference in accuracy between SVM (95%) and ANN (96%). Moreover, in [94], the researchers use ML for the classification of weeds and crops by using RF. The employed dataset is divided into different categories, specifically crop, weed, and irrelevant data. The authors train the model on offline datasets and apply these pre-trained models to real-time images. They have trained their system to give feedback to the flow control system. The RF algorithm gave 97% accuracy with a training time of 57.4 ms.

Moreover, in [98], the authors use RF for crop and weed classification through the following approach. They perform classification using NIR + RGB images, which were captured through a mobile robot. NIR can help distinguish the plant from the soil and background. This process is defined in four steps; firstly, identification of a plant using NIR information, which helps remove unrelated backgrounds so that only relevant regions can be considered for classification. Then masking is computed on pixel location. Secondly, feature selection has been performed on the relevant region. Then in the third step, RF is applied to those computed features, and a binary probability distribution is obtained, which described that the pixel belongs to a crop or a weed. In the fourth and last step, to improve the classification results, the information from the third step is utilized in Markov Random Field (MRF) by computing label assignment independently of the other nearby labels. In this way, authors were able to achieve 97% accuracy with RF.

In [99], the authors focus on identifying weeds from carrot fields to reduce the use of herbicides. During the development of plants, it is very difficult to discriminate between the color of a plant and weeds, which also makes the discrimination process even more difficult when both the plant and weed overlap each other. To address this problem, they proposed a 3-step procedure: (1) image segmentation. In this step, the input images are segregated from weeds using a normalization equation which gives higher weight to the greener part of the plants and removes the other colors from the input image, (2) in the second step, feature extraction is performed from the images got from the first step, and (3) in the third step, weed detection is performed through SVM algorithm. In addition, the overall accuracy obtained by SVM is 88%. In a related paper [100], the authors discussed the problem of overlapping weed and carrots leaves. In the initial stage of plant development, the color of both plant and weed are the same, which makes it more challenging to identify the weed and plant. Therefore, the 3 step procedure has been proposed to improve the detection or identification of plants and weeds. Initially, images are segmented using k-means clustering. Then, features are extracted from these segments by using HoG, which is then fed to SVM to acquire an improved accuracy of 92%.

In [103], the objective of this research is to propose a very accurate identification of weeds against crops using robots. The similarities between the shape of a plant and a weed make it challenging to identify plants precisely from weeds. For that reason, they tried to add different shapes to make a pattern for the individual range of the plants and tried to detect weeds based on these patterns using SVM and ANN to achieve maximum accuracies of 95% and 92%, respectively. Moreover, in [104], the authors compare the performance of several ML algorithms to detect the Canadian Thistle weed, particularly from a limited sample size of 30 images. The intent of the authors is also to demonstrate that, with the use of enhanced IP techniques, it can be possible to attain comparable performance with ML algorithms. Hence, the authors compare the performance of NBG, DT, KNN, SVM, and ANN algorithms with an IP technique in which they initially convert the image to grayscale, remove it from the green channel (in RGB), binarize it, and then perform morphological erosion to detect weed. In fact, this is not a new IP algorithm but rather a sequence of N/M IP techniques. The authors show that this IP method achieves comparable accuracy (98%) to the ML algorithms (97%, 96%, 96%, 96%, and 96%, respectively).

In [106], the authors have focused on developing a system that caters to the effect of using multiple image resolutions in the weed detection process. The authors employ enhancements of feature extraction, codebook learning (a clustering technique), feature encoding, and image classification as IP techniques. Particularly, the system takes an image as an input with 200 × 50 resolution, then feature extraction is performed by combining fisher encoding with codebook to cater to the limitation of feature extraction by using 2-level image representation. Then the image representation vectors got from feature extraction are given to the SVM algorithm for classification to achieve an overall accuracy of 89%. Finally, in [107], the authors focused on feature engineering, i.e., selecting the best set of features from gray-scale images by using HoG and LBP techniques. The extracted features are fed to two ML algorithms, i.e., ERT and RF, both of which give below-par accuracy of 52.5% and 52.4%, respectively, with a training time of 83 s and a limited customized dataset.

#### 7.1.3. DL.CNN Papers

In [33], the authors introduce the concept of positive (weed present) and negative (weed not present) images. They employ drone-acquired images of ‘black-grass’ and ‘common chickweed’ for the positive class and ‘wheat’, ‘maize’, and ‘sugar beet’ for the negative class. They pre-process images to avoid overfitting because of a small range of datasets and use the traditional (vanilla) CNN architecture with three combinations of convolution and max pooling layers to extract filters through the former and reduce size through the latter, followed by the one-dimensional fully-connected layer and a single output neuron for classification. The authors achieve an accuracy of 97%. Moreover, in [83], the authors employ transfer learning techniques to reuse the GoogleNet CNN that was previously trained on IARA datasets to classify three types of weeds, namely littleseed canarygrass, crowfoot, and jungle rice. The authors achieve an average accuracy of 98% across these three weeds.

In [85], the authors detect weeds from images of carrot fields to enhance the performance of an existing CNN architecture (with one convolution and max pooling layer only) through the use of GPUs. Although the accuracy remains exactly the same, the authors can attain a maximum speed-up of 2.0× (976 min on GPU as compared to 1895 min on CPU). In another application [31], the authors propose using CNNs to localize and classify weeds simultaneously from carrot field images acquired through robots to replace their current lengthy solution of multi-stage weed detection process through image segmentation. They experiment with both YOLO and GoogleNet to acquire a weed detection accuracy of 89% and 86%, respectively, which is a significant performance improvement over their image segmentation framework.

In [88], the researcher has used Mask R CNN for enhancement of accuracy in weed detection for the following weeds: mayweed, chickweed, blackgrass, shepherd’s purse, cleaver, fat-hen, and loose silky-bent. They employ Mask R CNN also for the segmentation of weed images. In both applications, Mask R performs better than FCN through a 100% accuracy in training and greater than 90% in the validation phase. In another application [89], the authors compare the performance of CNN with the HoG image processing method for weed detection. CNN application is conducted on hyperspectral images with four convolutional layers, two fully-connected ones, while RGB images are used with the HoG method. The results show that CNN can extract more discriminative features than HoG and with better accuracy (88%), although the computational processing required by CNN increases with the number of color bands.

Yet another comparison between CNN and IP techniques is done in [91], in which the authors develop a low-cost weed identification system that employs CNN. In the system, the data are initially collected and processed. Then, a relevant set of images is sampled, followed by weed detection through CNN. The authors also employ HOG and LBP approaches and achieve the best accuracy of 96% by initially employing LBP to extract relevant features and then using them as input to CNN. In [30], the authors generate synthetic datasets for weed classification based on real datasets by randomizing different features such as species, soil type, and light conditions. They compare the performance of weed detection over both synthetic and real datasets by using Segnet and Segnet-Basic CNNs and show that there is no performance degradation with synthetic datasets with the accuracy of 84% and 98%, respectively.

In [96], the authors indicate the limitations of detecting weeds with real-life images in that whole image content has to be fed into deep learning architectures, which sometimes makes it difficult to distinguish weeds from their background like soil. Hence, the authors propose using pre-trained deep learning models, particularly ResNet-50 for classification and YOLO for performance speed-up to achieve an accuracy of 99%. The authors create a framework to utilize both these models for weed detection. In a related work [81], the authors experiment with three different deep CNN architectures for weed detection, namely, DetectNet, GoogleNet, and VGGNet. They discovered that, for different types of active turfgrass weeds, VGGNet demonstrated much superior performance as compared to GoogleNet in different surface conditions, mowing heights, and surface densities. Moreover, DetectNet outperformed GoogleNet for dormant turfgrass weeds. The authors also demonstrate that image classification is an easier solution for weed detection as compared to object detection because the latter requires the use of bounding boxes.

In [101], the authors solve the tedious process of manually labeling image data at the pixel level by proposing a 2-step manual labeling process. Here, the first step is the segregation of foreground and background layers using maximum likelihood classification, with manual labeling of segmented pixels of background occurring in the second step. This setting can be used to train segmentation models which can discriminate between crops and other types of vegetation. The authors experiment with this approach using a SegNet model based on ResNet-50 and VGGNet encoder blocks, and UNet. The ResNet-50 SegNet model can demonstrate the best result (99%). Furthermore, in [105], the authors employ the AlexNet CNN architecture for weed classification in the ecological irrigation domain by using three different combinations of weeds and crops as datasets, with both CPU and GPU computing. They demonstrate a maximum accuracy of 99.89%. The authors validate that through their AlexNet application, both multiple and single weeds can be detected simultaneously, hence allowing enhanced irrigation control and management.

In [108], the authors developed intelligent software that is able to perform weed detection on-the-fly on multi-spectral RGB + NIR images acquired from the BOSCH Bonirob farm robot. For this, a lightweight CNN is initially used to extract pixels that represent projections of three-dimensional points belonging to green areas or vegetation. Then, a much deeper CNN uses these pixels to discriminate between crops and weeds. The authors also propose a novel data summarization method that selects relevant subsets of data that are able to approximate the original complete data in an unsupervised manner. The authors are able to achieve a maximum mean average precision (mAP) of 95%. A similar work is done in [110], where the authors use GoogleNet to detect weeds in the presence of a large amount of leaf occlusion. The loss function is guided by the bounding boxes and coverage maps of 17,000 original images collected from a high-speed camera mounted on an all-terrain vehicle. The authors manually annotate these images (which is a time-consuming activity) to achieve a precision of 86%, although the recall performance is poor (46%).

In [80], the author experiments with three CNN architectures, namely VGGNet, GoogLeNet, and DetectNet, for the recognition of broadleaf weeds in turfgrass areas. Through different experiments, the authors show that VGGNet demonstrates the best performance in classifying several different broadleaf weeds, while DetectNet outperformed the others in detecting one particular broadleaf weed. Furthermore, in [111], the authors sought to categorize the weeds in aerial photographs obtained from a height of under ten meters. The photos were taken using a 3024 × 4032 pixel resolution. Images were captured at the Heidfeldhof estate near Stuttgart’s Plieningen. Using a mobile, pictures were captured vertically at a height of 50 cm. The captured weed was in its early stages of development, and [135] weed photos were utilized to evaluate the model using pixel-based techniques. They use the CNN model and proposed two approaches, one is object detection, and the second is pixel-wise labeling. The object-based approach was applied to three different datasets, and the highest mAP achieved by this approach was 84.2%, and the pixel-wise approach achieved 77.6% as the highest mean accuracy using FCN.

In [114], the authors combine DL with IP for the classification of crops and weeds. Initially, a previously-trained CenterNet is used for detecting crops and drawing bounding boxes around them. Then, green objects falling outside these boxes are considered to be weeds, and the user can then focus only on crop detection with the reduced number of training images and easier weed detection. Moreover, the authors employ a segmentation-based IP method based on color indexing to facilitate the aforementioned detection of weeds, with the color index being determined through Genetic Algorithm optimization. This setup achieved a maximum precision of 95% for weed detection in crop/vegetable plantations.

In [116], the authors simply propose a framework for crop and weed classification using deep learning in real-time. They use Dicot and Monocot weeds. Images are being captured using a USB camera and processing of images has been done by using the OpenCV library. For weed classification, SSD objection detection is used, which uses a pre-trained VGG16 for mapping features from images and convolutional filter layers for the detection of weed. For three different settings, i.e., when the weeds and crops are overlapping and the weed size is smaller and larger than the crop size, the authors are able to acquire an average weed detection accuracy of 20% only.

In [117], the authors employ graph-based DL architecture for weed detection from RGB images which are collected from a diverse number of geographical locations, as compared to related works carried out in a controlled environment. Initially, a multi-scale graph is constructed over the weed image with sub-patches of different measures. Then, relevant patch-level patterns are selected by applying a graph pooling layer over the vertices. Finally, RNN architecture is used to predict weeds from a multi-scale graph with a maximum accuracy of 98.1%. In a related work [118], the authors use a feature-based GCN to detect weeds. They construct a GCN graph based on features extracted through CNN and the Euclidean distance between these features. This graph uses both labeled and unlabeled image features for semi-supervised training through information propagation and labeled data for testing. By combining GCN with ResNet-101, the authors were able to acquire accuracies of 97.80%, 99.37%, 98.93%, and 96.51%, respectively, on four different datasets, outperforming the following state-of-the-art methods: AlexNet, VGG16, and ResNet-101, with a reduced running time of 1.42 s.

In [119], the authors propose a semantic segmentation procedure for weed detection with ResNet-50 as the backbone architecture. They employ a particular type of convolution called hybrid dilation for increasing the receptive field and DropBlock for regularization through random dropping of weights. They also optimize RGB-NIR bands into RGB-NIR color indices to make the classification results more robust and employ an attention mechanism to focus the CNN on more correlated regions along with a spatial refinement block for fusing feature maps of differing sizes. The authors test their complicated approach on Bonn and Stuttgart datasets and compare the weed detection performance with UNet, SegNet, and FCN, along with performance over two other semantic segmentation algorithms, i.e., PSPNet and RSS [12]. For both datasets, they achieve better accuracy than the above five algorithms of 75.26% and 72.94%, respectively.

In [121], the authors employ the SSD to detect weeds in rice fields which employs VGG16 to extract features from images. Such a setting gives a maximum accuracy of 86% over different image resolutions, by using multi-scaled feature maps and convolution filters. The authors mention that the accuracy achieved with VGG16 (before re-usage) was 99%.

Finally, in [122], the authors employ the YOLOv3 CNN to discriminate between crops (sugar beet) and weeds (hedge bindweed). They use a combination of synthetic and real images and a K-means algorithm to estimate the anchor box sizes for YOLOv3. A test run on 100 images shows that synthetic images can improve the overall mean average precision (MAP) by more than 7%. The system is also able to demonstrate better performance and trade-off between accuracy and speed as compared to other YOLO variants.

Moreover [123], the researchers compared the performance of pre-trained classification algorithms such as VGG16, ResNet50, and Inceptionv3 for weed classification. Cocklebur, foxtail, redroot pigweed, and gigantic ragweed are four weeds commonly seen in corn and soybean fields in the Midwest of the United States. They also used YOLOv3 object detection to locate and classify weeds in an image dataset. VGG16 outperformed all pre-trained models with an accuracy of 98.90%. They also compare Keras with Pytorch, finding that Pytorch takes less time to train models and has higher accuracy than Keras.

The authors in [124] examined the performance of single shot detector (SSD) and Faster RCNN in terms of weed detection utilizing images of soybean fields recorded with a UAV in this study. Both the single shot detector and the quicker RCNN object detection algorithms were compared to the patch-based CNN model. According to the authors, Faster RCNN outperformed the SSD Model. Furthermore, faster RCNN outperformed patch-based CNN.

The authors of [125] research proposed a vision-based classification method for weed identification in spinach, beet, and bean. CNN was used for classification. UAV was used to capture the images used in this section. Precision was used to measure model performance, and beet received the highest precision of 93%. Additionally, The researchers in [126] attempted to construct a precision herbicide application using DCNN and its various variations such as VGGNet, DetectNet, GoogleNet, and AlexNet for the detection of various weeds, such as dandelion, ground ivy, and spotted spurge in this work.

To make the algorithms more manageable for hardware with low resources while still retaining accuracy, in this study [127] the authors used ensemble learning approaches, transfer learning, and model compression. The suggested method was carried out in three steps: transfer learning, pruning-based compression, quantization, and Huffman encoding, and model ensembling with a weighted average for improved accuracy. Similarly in [128], researchers presented a method for locating a specific area and applying herbicide based on object detection in real-time as well as crop and weed classification. In this study, two weed types—monocotyledon and dicotyledon—that are typically seen in cereal crops were specifically targeted. They acquired 1318 photos using a Nikon 7000 camera for field recording, trained CNN for classification under various lighting situations, and trained YOLO for object detection. This [129] research study offered a novel deep-learning technique to categorize weeds and vegetable crops. CenterNEt, YOLO-v3, and Faster RCNN were employed in this approach. The YOLO-v3 model was the most effective in identifying weeds in vegetable crops out of the three. For the pixel-by-pixel segmentation of weed, soil, and sugar beet, [130] the author employed ResNet50 and U-Net. For 1385 photos, they employed these models as encoder blocks, and to deal with unbalanced data, they also applied a unique linear loss function. CNN was primarily employed for the classification and spraying of certain areas for herbicide application. The segmentation accuracy in tiny regions was increased by using a bespoke loss function and balanced data.

#### 7.1.4. ML.DL.CNN Papers

In [28], the authors compare the performance of SVM, ANN, and CNN for discriminating between crops and weeds, specifically four different crop types and Paragrass and Nutsedge weed types. They employ median and Gaussian filters for identifying the relevant areas in images and also extract shape features for both crops and weeds. SVM is assessed over two kernel functions, i.e., radial basis and polynomial, while ANN is evaluated with one hidden layer containing six neurons, with the output layer containing two neurons (one each for weed and crop detection). The CNN contains the traditional convolutional and maxpooling layer (with ReLU activation) followed by the fully connected layer. The authors show that, in the best result, ANN is the best classifier for both weed and crop classes, followed by SVM and then CNN.

In [86], the authors use SVM and ResNet-18 classifier to discriminate between weeds and crops from unsupervised (unlabeled) images collected from a UAV. They extract deep features from the images and employ a one-class classification approach with the SVM classifier. Hough transform and SLIC are used to detect the crops’ rows and segment the images into superpixels, which are used to train the SVM. It is found that the performance of SVM is comparable with the performance of a ResNet-18 CNN which has been trained through supervised learning (maximum 90%).

In [87], the authors focus on broad-leaf weed detection in pasture fields through an application and comparison of both ML and DL algorithms, namely, SVM (with linear, quadratic, and Gaussian kernel), KNN, Ensemble subspace discriminant, Regression and CNN consisting of six convolutional layers and alternating max-pooling and drop-out layers and three fully connected layers. Local binary pattern histogram (LBPH) is used to extract information from grayscale and RGB images. The authors demonstrate that CNN outperforms all ML variants by giving a maximum accuracy of 96.88%.

In [102], the authors employ CaffeNet (a variant of AlexNet) for grass weed and broadleaf weed detection in soybean crop images captured from the Phantom DJI drone and compare its performance with SVM, Adaboost, and RF algorithms. SLIC was used to extract superpixels for input to all algorithms. Although CaffeNet achieved the best accuracy of 99%, SVM, Adaboost, and RF also achieved similar results with 97%, 96%, and 93% accuracy, respectively.

In [109], the authors address the particular problem of manually annotating and/or segmenting a large number of UAV/drone images for a supervised weed detection task. They propose an automated unsupervised method of weed detection based on CNNs. Initially, they detect crop rows using Hough transform variations and SLIC. The output is a set of lines identifying the center of the crop rows, i.e., around which the crops are growing. Applying a blob-coloring algorithm on these lines to represent the crop regions, anything that falls outside the blob area (crop vegetation) is a potential weed. These weeds are then labeled autonomously and form the dataset for CNN, i.e., ResNet-18. In the data of bean fields, the best accuracy is obtained by ResNet (88.73%), followed by RF (65.4%) and SVM (59.51%), while for the spinach field dataset, RF is the winner with 96.2% accuracy, followed by ResNet-18 (94.34%) and SVM (90.77%).

Moreover, a thorough comparison between ANN and AlexNet CNN has been done by the authors in [115], in which they develop an application to transmit drone-captured images to a machine learning server. The results demonstrate that AlexNet is able to acquire a maximum accuracy of 99.8% while the maximum achieved by ANN is only 48.09%.

In [120], the authors attempt to construct an automated weed detection system that can detect weeds in their different stages of growth and soil conditions. For this, they employ a set of pre-trained CNN architectures, namely Inception-Resnet, VGGNet, MobileNet, DenseNet, and Xception, through transfer learning techniques to extract deep features. Then, each of these feature sets is used for weed classification with a set of traditional ML algorithms, specifically, SVM, XGBoost, and LR. The authors test the system on tomato and cotton fields over black nightshade and velvetleaf weeds. The authors claim that the best F1 score of 99.29% is achieved by Densenet and SVM, while all other CNN-ML combinations give an F1 score greater than 95%.

#### 7.1.5. AE Papers

In [92], the authors focus on the problem of designing an automated weed detection system that can generalize to varying environments and soil conditions, as well as weed and crop types. For this, they propose an autoencoder architecture, embedded within an FCN, which generates two types of features through the downsample-upsample process. First are visual features that are generated for each image through the visual code generated after downsampling, and the second are sequential features that are generated through a sequence code that aggregates data from a batch of images acquired from the Bonirob robot. Both visual and sequence features are combined into a pixel-level label mask that is able to distinguish between both crops and weeds distinctly. In comparison to a baseline method and some previous approaches, the proposed approach demonstrates better precision and recall for both crops and weeds over the Stuttgart and Bonn datasets.

In another paper by the same research group [93], the authors use a similar approach to identify the actual stems of the weeds for mechanical control (e.g., pulling out) and also a surrounding region for effective spraying. For this, they initially generate a visual code for each image, which is then input to two different decoder networks, specifically, one which outputs a pixel map related to weed stem detection and the other for crop detection. This information is used to identify a bounding area around the stems for spraying. The authors show that their system can achieve better average precision for identifying two types of weeds (dicot and grass weed) than a baseline and other related systems.

In [97], the authors employ two variants of the SegNet algorithm (SegNet 512 and SegNet 256) to detect weeds from the CWFID dataset. They also make several architectural changes to the original SegNet architecture to enhance the downsampling performance for both SegNet 512 and SegNet 256, for instance, by adding or removing convolution and batch normalization layers, changing the kernel size and the size of the hidden layers. As the focus of the authors here is on the decoder’s performance, we have categorized this paper under the DL.AE label. The validation and test accuracy over SegNet 512 is 92% and 96% respectively, while for SegNet 256, the corresponding accuracy is 92% and 93% respectively. The authors also show that the training, evaluation, and prediction time for SegNet 512 is understandably twice as much for SegNet 256 because the former employs twice as more upsampling and downsampling blocks as compared to the latter.

In [112], the authors conduct a performance comparison of DeepLab V3, U-Net, and SegNet, which are all autoencoder-based CNN variants. Initially, patches or relevant regions are selected from aerial images of sugar beet crops to generate the relevant set of features, which are then input into the three variants. The results show that DeepLab V3 demonstrates the best AUC values for both crop and weed identification, while U-Net performs better than SegNet. However, DeepLab V3 is computationally the most expensive, followed by SegNet and then U-Net. The authors recommend generating smaller patches over a larger training data size with an application of U-Net to achieve a balance of speed and efficiency.

In [113], the authors employ the encoder-decoder architecture for semantic segmentation of weeds and crops. The encoder employs Atrous convolution (similar to DeepLab) over four convolution layers and one pooling layer with an output code of size 1X1. This code is then upsampled in the decoder twice with several low-level features (from the atrous convolution output of the encoder) as input. Different image enhancement techniques were compared and used for improving the quality of images and for making the model to be robust against different lighting conditions. The results demonstrate that when NIR color indices are used with these enhancement techniques, the weed identification performance is significantly improved. However, without NIR indices, pure image enhancement techniques demonstrate an average performance even though they still improve the quality of images under different lighting conditions.

## 8. Challenges and Future Research Directions

In this section, we answer RQ4: What are the tangible future research directions to achieve further benefit from deep learning applications for weed detection? For this, we identify and divide the directions of future research and challenges in the domain of deep learning applications for weed detection into two parts: domain and technical.

### 8.1. Domain Challenges

**Missing integrated image databases:** There is a need to create a general repository of weed image datasets with specified associations to their respective crops, generated with high-speed cameras (either mounted on UAVs/robots or taken manually), of an agreed-upon high resolution, and categorized according to different modality types. This will create proper benchmarks for any future weed detection experiments. For instance, an experiment to detect Canadian thistles in some European countries can employ the standard Canadian Thistle images as the baseline. The need arises from the fact that almost all researchers generate their own datasets using different types of cameras without any baseline images, which makes it difficult to determine the exact impact of their work on the research community.**Lack of standards:** The main challenge arising from implementing such a standard repository is that weeds demonstrate significant diversity from each other, as do their associated crops. Both weeds and crops can demonstrate different growth conditions (size, density, etc.) with respect to weather and other external variables, and the effects of shadows and illumination will require further classification (and hence complexity) of the resulting images. Moreover, manually annotating each image separately over different classifications is a complex task. Catering to all of these requirements in collecting weed and their associated crop images is a challenging task.**Environmental Challenges:** The environmental indicators such as soil temperature, soil water potential, exposure to light, fluctuating temperatures, nitrates concentration, soil PH and the gaseous environmental soils impact the composition of weed flora of the cultivated area. Therefore, it is essential to understand the usage of soil profiling and temperate can help predict early weed detection. However, creating a soil profile is a time-consuming task because of the nature of the soil variant.

### 8.2. Technical-Related

We believe that our analysis of Table 7 provides a clear roadmap for practitioners to derive multiple lines of future research with respect to the selection of algorithm to detect a particular type of weed and/or associated crops or selection of weeds/associated crops for detection. Moreover, from an algorithmic perspective, it is obvious that DL, particularly with the use of CNN and its variants, has the power to generate satisfactory predictive performance for weed detection. As this trend is prevalent and rising, we expect it to continue in the near future. As more variants of CNNs are discovered, there is a high probability that they will be soon applied for weed detection and its related field. In our opinion, the distinction between the performance of ML and DL can only be clarified after thorough experimentation of CNN/variants with more robust ML models, particularly, SVM, Boosting variants (Adaboost, XGBoost, LightGBM), LR, and RF, over different standard weed datasets. Although we have discussed such previous applications, they do not demonstrate clearly that DL has a significant edge over ML applications, and hence, these results cannot be considered comprehensive and generalizable in our opinion.

Furthermore, once we have some standard baseline repository of images as proposed above, we propose an application of CNN on these baseline datasets to provide an actual benchmark performance over different measures, specifically accuracy, precision, recall, F1-score, and AUC score. The reason is that researchers are now focusing on improving CNN’s performance further through the use of different variants, notably ResNet, VGGNet, and SegNet. As this trend is increasing rapidly, we expect it to continue. Our proposed baseline performance benchmarks will then provide a standard backbone to compare the performance of any application of CNN variant over any weed type and to position the paper with respect to its comparison with proposed baselines. In doing so, one can also try to address the prevalent problems of natural light variation and weather effects.

Another research direction is to quantify the impact of using standard and well-known IP techniques for both DL and ML algorithm applications, particularly feature selection, BFS, image segmentation, cluster analysis, and different transformations, for the exact problem of distinguishing the weeds from their respective crops in the same image. Moreover, we have seen that very good results have been acquired in both DL and ML applications without the use of any IP technique. So, there is a need to understand, quantify and hence standardize the impact of these techniques for crop-weed discrimination and a generalized perspective.

Moreover, it remains to be explored how ML and DL applications are impacting related fields such as pest and disease detection and the impact of transfer learning of CNN-based models from one domain to the other. Finally, we believe there is a need to design appropriate software architectures for such a weed detection activity which could be generalized for future applications.

## 9. Conclusions

This paper conducts the first SLR to review deep learning applications in depth for weed detection. We adopt the standard SLR methodology and answer four concrete research questions to thoroughly summarize the state-of-the-art research’s impact and articulate domain and technical challenges for future research directions. Furthermore, we created a citation graph to understand the pattern of publications and researchers in this area. We also compare our work with the eight latest literature reviews and demonstrate our approach’s superiority and differences with these reviews.

## Figures and Tables

**Figure 1 sensors-23-03670-f001:**
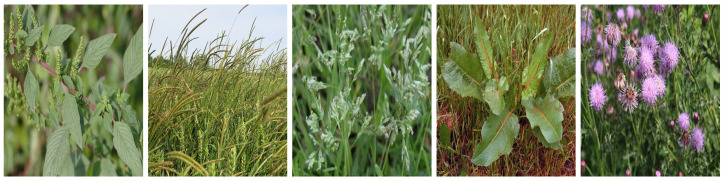
From left to right: Pigweed, Blackgrass, Bluegrass, Dockleaf, Canadian Thistle (Source: [41]).

**Figure 2 sensors-23-03670-f002:**
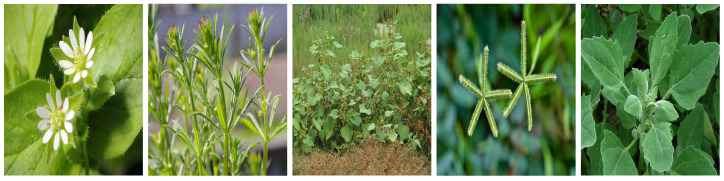
From left to right: Chickweed, Cleaver, Cockleblur, Crowfoot, Fat-hen (Source: [41]).

**Figure 3 sensors-23-03670-f003:**
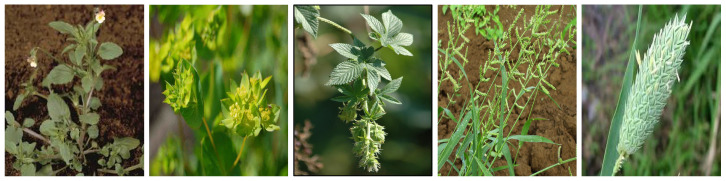
From left to right: Field pansy, Hare’s ear mustard, Japanese hop, Jungle rice, Little seed (Source: [41]).

**Figure 4 sensors-23-03670-f004:**
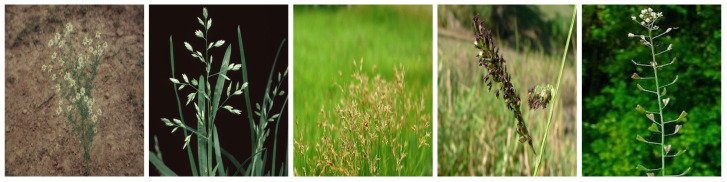
From left to right: Mayweed, Meadow grass, Nutsedge, Paragrass, Shepherd’s purse (Source: [41]).

**Figure 5 sensors-23-03670-f005:**
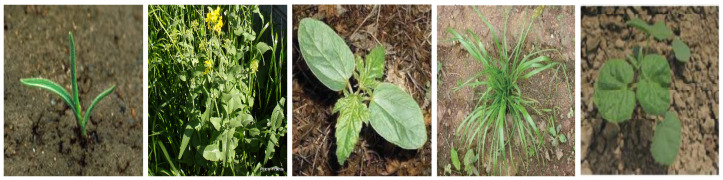
From left to right: Silky-bent, Turnip weed, Dicot, Grass Weed, Velvetleaf (Source: [41]).

**Figure 6 sensors-23-03670-f006:**
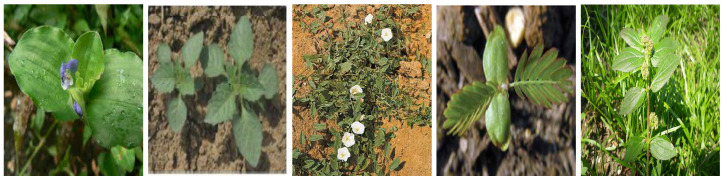
From left to right: Benghal dayflower, black nightshade, hedge bindweed, Indian jointvetch, snakeweed (Source: [41]).

**Figure 7 sensors-23-03670-f007:**
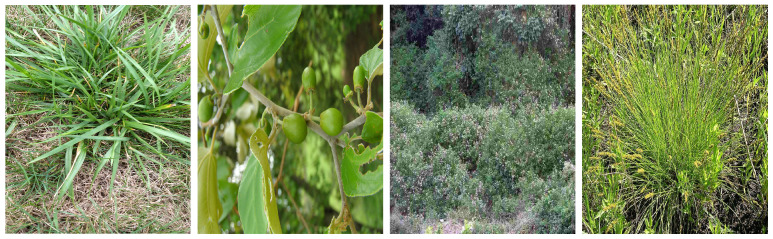
From left to right: Fescue grass, Chinee apple, *Lantana camara*, Sedge weed (Source: [41]).

**Figure 8 sensors-23-03670-f008:**
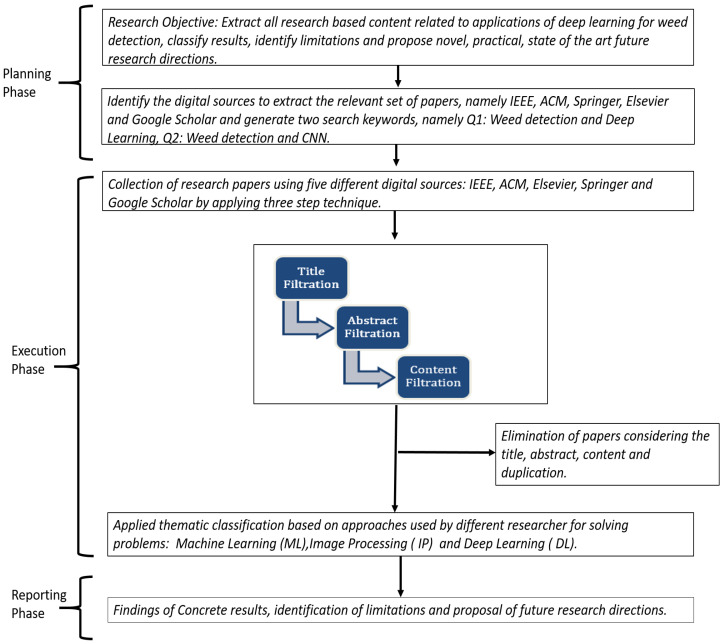
Our SLR process.

**Figure 9 sensors-23-03670-f009:**
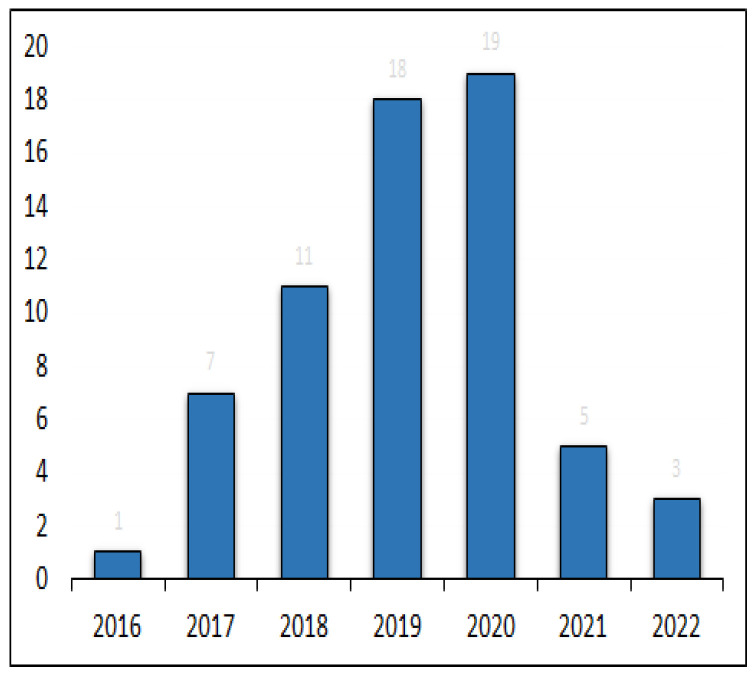
Year Wise Distribution of Articles.

**Figure 10 sensors-23-03670-f010:**
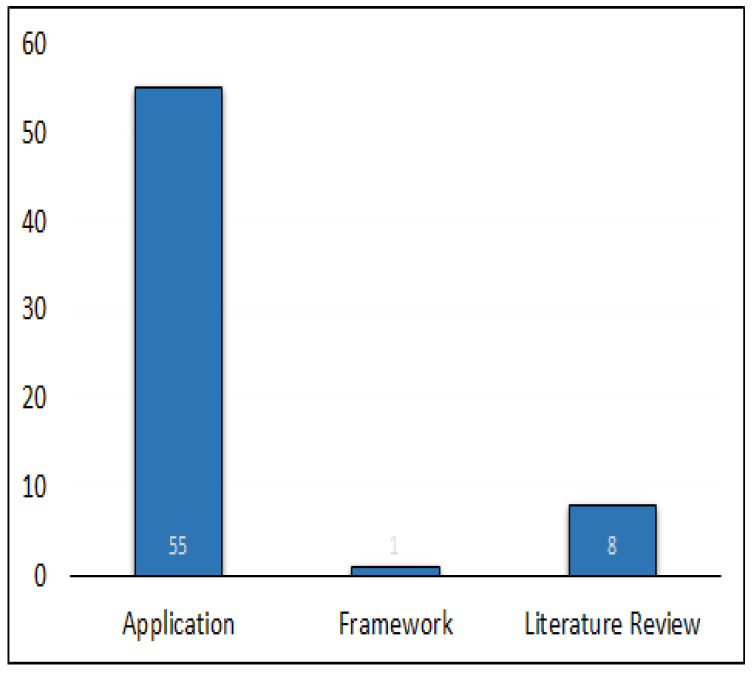
Article types and Frequency.

**Figure 11 sensors-23-03670-f011:**
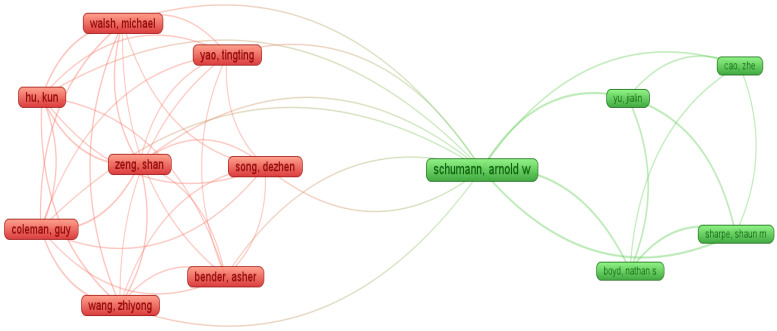
Co-Author Citation Graph.

**Figure 12 sensors-23-03670-f012:**
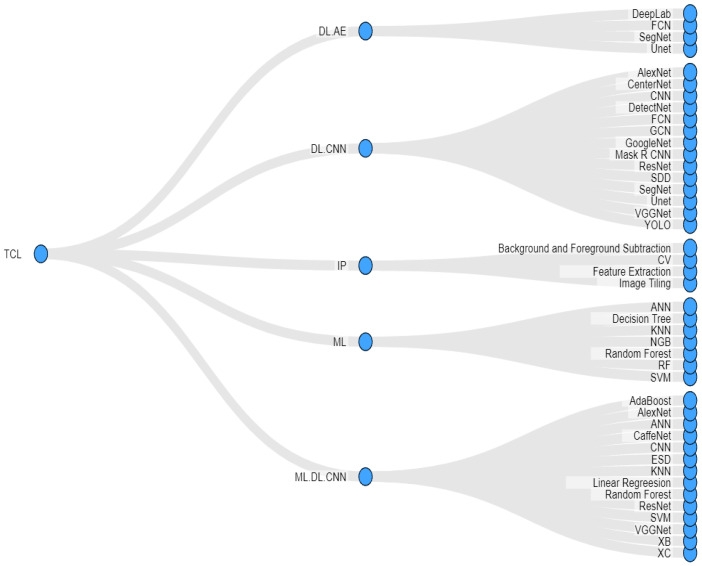
The distribution of different algorithms with respect to five thematic classification labels (TCLs).

**Figure 13 sensors-23-03670-f013:**
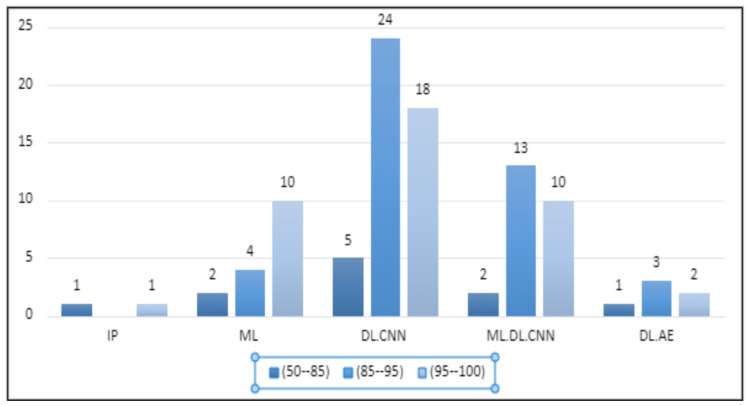
Observed performance evaluation with respect to our defined thematic classification labels.

**Table 1 sensors-23-03670-t001:** Breakdown of frequency of articles filtered with respect to title, abstract, and content across search queries Q1. Q2 and Q3 and digital sources.

	Q1			Q2			Q3		
**Digital Sources**	**Title**	**Abstract**	**Content**	**Title**	**Abstract**	**Content**	**Title**	**Abstract**	**Content**
IEEE	50	42	25	4	4	4	0	0	0
ACM	3	2	1	2	2	1	0	0	0
Elsevier	8	7	8	3	3	2	0	0	0
Springer	8	7	6	3	3	2	0	0	0
Google Scholar	10	5	9	9	2	2	11	6	4

**Table 2 sensors-23-03670-t002:** Information regarding weed datasets used in application papers.

Article	Dataset	Size	Crop	Weed	Modality	Data Collection	Resolution
[32]	Gharo	N/M	Wheat	N/M	N/M	Drone	4000 × 3000
[33]	Sensifly	1500	Wheat, Maize, Sugar beet	Chickweed, Blackgrass	RGB	Referred Dataset	150 × 150
[83]	IARI	60	Rice, Maize	Littleseed canarygrass, Crowfoot, Jungle rice	N/M	Referred Dataset	800 × 450
[84]	Tegucigalpa	N/M	Vegetables	N/M	RGB	Semi-Professional Camera	4512 × 3000
[85]	Carrot Field	N/M	Carrot	N/M	N/M	Camera	N/M
[28]	Tumakuru	2560	Chrysanthemum	Paragrass, Nutsedge	RGB	Digital Camera	250 × 250
[31]	Heide Weed	2907	Carrot	N/M	RGB	Robot(BoniRob)	832 × 832, 416 × 416 and 288 × 288
[86]	WeedMap	11,441	Sugar beet	N/M	RGB, CIR	Drone	480 × 360
[87]	Dundee	6087	Pasture	Dockleaf	RGB, Greyscale	Camera	64 × 64
[88]	Plant Seedling	5539	Sugar beet, Maize, Wheat	Mayweed, Chickweed, Blackgrass, Shepherd’s purse, Cleaver, Fat-hen, Loose silky-bent	N/M	Referred Dataset	N/M
[89]	Brimrose	N/M	N/M	N/M	Hyperspectral	Brimrose VA210, JAI BM-141 camera	500 × 500, 250 × 250, and 125 × 125
[90]	Campo Grande	N/M	Soybean	Dockleaf	RGB	Drone	4000 × 3000
[91]	Luye	N/M	N/M	N/M	RGB	Drone	1920 × 1080
[30]	Bosch BoniRob	1300	Sugar beet	Shepherd’s purse	RGB + NIR	Robot	1296 × 966
[82]	Cherry Research Farm	N/M	Strawberry	Fescue grass	RGB	Referred Dataset	N/M
[92]	Bosch BoniRob, Stuttgart	N/M	Sugar beet	N/M	RGB + NIR	Robot	N/M
[93]	DeepField, Bosch BoniRob	N/M	Sugar beet	Dicot, grassweed	RGB, RGB + NIR	Drone, Robot	N/M
[94]	Crop Weed	291	N/M	N/M	RGB	Camera	N/M
[95]	CWFID	N/M	Carrot	N/M	RGB (Multispectral)	Robot	N/M
[96]	CropDeep	1000	Corn	Bluegrass, Fat-hen, Canadian thistle, Sedge weed	N/M	Referred Dataset	224 × 224
[97]	CWFID	120	N/M	N/M	RGB (Multispectral)	Referred Dataset	N/M
[98]	Bosch BoniRob	N/M	Sugar beet	N/M	RGB + NIR	Robot	1296 × 966
[99]	CWFID	60	Carrot	N/M	RGB (Multispectral)	Referred Dataset	N/M
[100]	CWFID	60	Carrot	Mayweed	RGB (Multispectral)	Referred Dataset	N/M
[81]	Golf Field	36,000	Turfgrass	Dockleaf	N/M	Camera	640 × 360
[101]	Manitoba	906	Canola	N/M	RGB	Camera	1440 × 960
[102]	Campo Grande	400	Soybean	Dockleaf	RGB	Drone	4000 × 3000
[103]	Shiraz University	N/M	Sugar beet	Pigweed, Fat-hen, Hare’s-ear mustard, Turnip weed	RGB	Camera	960 × 1280
[104]	ImageNet	N/M	Cereal	Canadian thistle	RGB	Canon PowerShot G15 Camera	200 × 200
[105]	Ecology	3266	Maize, Wheat, Peanut Seedlings	Fat-hen, Japanese hop, Cocklebur	N/M	Drone	227 × 227
[106]	Rumex 100	900	Grass	Dockleaf	N/M	Referred Dataset	200 × 150
[107]	Tobacco Field	76	Tobacco Seedling	N/M	N/M	Camera	65 × 65
[108]	Bosch BoniRob	900	Sugar beet	N/M	RGB + NIR (Multispectral)	Robot	61×61
[109]	Bean, Spinach	5534	Bean, Spinach	N/M	RGB	Drone	64 × 64
[110]	Cereal	N/M	Cereal	N/M	N/M	Camera	1224 × 1024
[80]	Griffin	4550	Turfgrass, Bahigrass	Dockleaf	RGB	Digital Camera	640 × 256
[111]	Heide Feldhof Farm	796	Wheat	Mayweed, Meadowgrass, Chickweed, Field pansy, Pigweed	N/M	Terrestrial Images (Cellphone Camera)	N/M
[112]	DeepLab	N/M	Sugar beet	N/M	RGB, NIR, CIR, NDVI	Drone	480 × 360
[113]	Bosch BoniRob, OilSeed	280	Sugar beet	N/M	RGB, NIR	Robot, Camera	1296 × 966
[114]	Bok Choy	11,150	Chinese White Cabbage	N/M	RGB	Camera	512 × 512
[115]	Reduit	15,336	Soybean	Dockleaf	N/M	Drone	227 × 227.
[116]	MonoDicot	N/M	Ragi	Grass weed, Dicot	N/M	Camera	N/M
[117]	DeepWeeds	17,509	N/M	Chinee apple, Snake weed, Lantana	RGB	Referred Dataset	256 × 256
[118]	Radish Weed	6000	Corn, Lettuce, Radish	Nutsedge, Fat-hen, Canadian thistle	RGB	Camera	800 × 600
[119]	Bonn, Stuttgart	9070	Sugar beet	N/M	RGB-NIR	Camera	512 × 384
[120]	Greece Farm	504	Tomato, Cotton	Black nightshade, Velvetleaf	RGB	Camera	128 × 128
[121]	FUT Farm	5400	Rice	Indian jointvetch, Benghal dayflower, Jungle rice	N/M	Camera	250 × 250
[122]	Flanders	652	Sugar beet	Hedge Bindweed	RGB	Camera	800 × 1200
[123]	Annotaed Imagery Dataset	462	Corn, Soybean	Cocklebur, foxtail, redroot pigweed and giant ragweed	RGB	Sony WX350, Panasonic DMC-ZS50	1200 × 900
[124]	UAV Imagery	N/M	Soybean	N/M	N/M	UAV	N/M
[125]	UAV Imagery	N/M	Spinach, beet, Bean	N/M	N/M	UAV	7360 × 4972
[126]	E.maculata,	N/M	N/M	dandelion, ground ivy, spotted spurge.	N/M	UAV	426 × 240
[127]	Giselsson	5539	N/M	N/M	N/M	Referred Dataset	N/M
[128]	Beni Mellal-Khenifra	1318	Cereal crops.	Monocotyledon and dicotyledon	N/M	Nikon 7000 camera	N/M
[129]	Dataset	9200	Vegetables	N/M	N/M	digital camera	2048 × 1536
[130]	Dataset	1385	Sugar beet	N/M	RGB	digital camera	480 × 640

**Table 3 sensors-23-03670-t003:** Algorithmic Performance and Related Data for our application papers.

Article	Algorithm	IPT	KPI	Result	TR.Time	TR.TS.Split	TCL
[32]	N/M	Background Subtraction	Accuracy	67%	N/M	N/M	IP
[33]	CNN	N/A	Accuracy	97%	N/M	72–27%	DL.CNN
[83]	GoogleNet	N/A	Accuracy	98%	N/M	83–16%	DL.CNN
[84]	N/M	CV (rgb2gray, im2bw, bwlabel, regionprops)	Accuracy	99%	2.98 s	N/M	IP
[85]	CNN	N/A	PR, RC	91.1%, 86.8%	1895 min (CPU), 976 min (GPU)	N/M	DL.CNN
[28]	SVM, ANN, CNN	Median and Gaussian filter	Accuracy	87%, 93%, 98%	N/M	90–10%	ML.DL.CNN
[31]	YOLO, GoogLeNet	N/A	Accuracy	89%, 86%	N/M	90–10%	DL.CNN
[86]	ResNet18, SVM	Hough transform, SLIC	Accuracy	90%	N/M	N/M	ML.DL.CNN
[87]	SVM, KNN, Ensemble Subspace Discriminant, CNN	CV	Accuracy	89%, 84%, 87%, 93.15%	N/M	80–20%	ML.DL.CNN
[88]	Mask R CNN, FCN	N/A	Accuracy	>90%, <90%	N/M	60–40%	DL.CNN
[89]	CNN	HoG	Accuracy	88% (CNN)	N/M	60–40%	DL.CNN
[90]	SVM, ANN	SLIC	Accuracy	95%, 96%	0.0211	60–40%	ML
[91]	CNN	LBP, HoG	Accuracy	96% (CNN)	N/M	N/M	DL.CNN
[30]	SegNet, SegNet-Basic	N/A	Accuracy	84%, 98%	0.14 s, 0.08s	N/M	DL.CNN
[82]	N/M	Feature Extraction, Classification	N/M	N/M	N/M	N/M	IP
[92]	FCN	N/A	PR, RC	97.9%, 87.8%	N/M	N/M	DL.AE
[93]	FCN	N/A	Avg. PR	87.90%	N/M	N/M	DL.AE
[94]	RF	N/A	Accuracy	97%	57.4 ms	75–25%	ML
[95]	N/M	Foreground Extraction, Image Tiling, Moment-Invariant Feature Extraction	N/M	N/M	480s	N/M	IP
[96]	ResNet-50, YOLO	N/A	Accuracy	99%	N/M	80–20%	DL.CNN
[97]	SegNet256, SegNet512	N/A	Accuracy	96%	11,389 s	75–25%	DL.AE
[98]	RF	Background Subtraction, Masking, Feature Extraction, Markov Random Field	PR, RC	95%, 96%	N/M	N/M	ML
[99]	SVM	Image Segmentation, Feature Extraction	Accuracy	88.99%	N/M	70–30%	ML
[100]	SVM	Image Segmentation (K-Means), Feature Extraction (HoG)	Accuracy	92%	N/M	70–30%	ML
[81]	DetectNet, GoogLeNet VGGNet	N/A	Accuracy	99%, 50%, 90%	N/M	78–22%	DL.CNN
[101]	SegNet, VGGNet, U-NET	Background Subtraction, Image Labeling,	Accuracy	99% (SegNet), 96% (VGGNet), 97% (U-Net)	N/M	85–15%	DL.CNN
[102]	CaffeNet, SVM, AdaBoost, RF	N/A	Accuracy	99%, 97%, 96%, 93%	N/M	75–25%	ML.DL.CNN
[103]	SVM, ANN	N/A	Accuracy	95%, 92%	N/M	60–40%	ML
[104]	NBG, DT, KNN, SVM, ANN	Morphological Erosion	Accuracy	97%, 96%, 96%, 96%, 96%, 98% (IP),	N/M	N/M	ML
[105]	AlexNet	N/A	Accuracy	99.89%	468 s with double GPUs	70–30%	DL.CNN
[106]	SVM	Feature Extraction, Codebook Learning (Clustering), Feature Encoding	Accuracy	89%	N/M	50–50%	ML
[107]	ERT, RF	LBP, HOG	Accuracy	52.5%, 52.4%	83 s	68–32%	ML
[108]	CNN	N/A	mAP	95%	3.6–4.5 s	N/M	DL.CNN
[109]	ResNet18, SVM, RF	N/A	AUC	95%, 95%, 97%	N/M	80–20%	ML.DL.CNN
[110]	GoogleNet	N/A	PR, RC	86%, 46%	N/M	N/M	DL.CNN
[80]	VGGNet, GoogLeNet, DetectNet	N/A	Accuracy	99% (VGGNet)	N/M	88–12%	DL.CNN
[111]	VGGNet, CNN	N/A	mAP	84%	N/M	80–20%	DL.CNN
[112]	SegNet, U-Net, DeepLabV3	N/A	AUC, F1-score	85%, 72%, 92%, 85%, 97%, 92%	N/M	N/M	DL.AE
[113]	DeepLab	Histogram Equalization	MIoU	96%	N/M	70–30%	DL.AE
[114]	CenterNet	Background Subtraction, Image Segmentation	PR	95%	N/M	80%–20%	DL.CNN
[115]	AlexNet, ANN	SLIC	Accuracy	99%, 48.09%	N/M	70–30%	ML.DL.CNN
[116]	SSD, VGGNet	CV	N/M	N/M	N/M	80–20%	DL.CNN
[117]	CNN	Graph Feature Extraction	Accuracy	98%	N/M	80–20%	DL.CNN
[118]	GCN-ResNet101, AlexNet, VGGNet, ResNet101	N/A	Accuracy	98% (GCN-ResNet101)	1.42 s	70–30%	DL.CNN
[119]	ResNet50, U-Net, SegNet, FCN	Image Segmentation (PSPNet, RSS)	mAP	93% (ResNet50)	N/M	N/M	DL.CNN
[120]	Inception-Resnet, VGGNet, MobileNet, DenseNet, Xception, SVM, XGBoost, LR	N/A	F1-score	99% (DenseNet, SVM)	N/M	N/M	ML.DL.CNN
[121]	SSD	N/A	Accuracy	86%	N/M	90–10%	DL.CNN
[122]	YOLO	K-Means	mAP	89%	6.48 ms	85–15%	DL.CNN
[123]	VGG16, ResNet50, Inception30, YOLOv3	N/A	accuracy	98%	354 s	60–40%	DL.ML.CNN
[124]	SSD, Faster RCNN, CNN	N/A	Precision	65%	N/M	N/M	DL.ML.CNN
[125]	CNN	N/A	Precision	93%	N/M	N/M	DL.ML.CNN
[126]	DCNN	N/A	F1-score	92%	N/M	N/M	DL.ML.CNN
[127]	VGG16, ResNet50, DenseNet	N/A	accuracy	91%	N/M	N/M	DL.ML.CNN
[128]	YOLO	N/A	accuracy	83%	N/M	N/M	DL.ML.CNN
[129]	YOLO-v3, CenterNet, and Faster R-CNN	N/A	F1-score	97%	N/M	N/M	DL.ML.CNN
[130]	U-Net, ResNet	N/A	IoU	96%	N/M	N/M	DL.ML.CNN

**Table 4 sensors-23-03670-t004:** Frequency Distribution of Algorithms.

Algorithm	Frequency
CNN	65
SVM	12
RF	5
ANN	5
KNN	2
Boosting	2
DT	2
NBG	1
Linear Regression	1

**Table 5 sensors-23-03670-t005:** Frequency Distribution of CNN Variants.

DL Algorithm	Frequency
CNN	11
ResNet	8
SegNet	7
VGGNet	8
GoogleNet	5
FCN	4
YOLO	4
U-Net	3
AlexNet	3
DetectNet	2
DeepLab	2
SSD	3
Mask R CNN	2
CaffeNet	1
CenterNet	1
GCN	1
MobileNet	1
DenseNet	1
Xception	2

**Table 6 sensors-23-03670-t006:** Frequency Distribution of IP Techniques.

IP Techniques	Frequency
Feature Extraction	6
Background and Foreground Subtraction	5
Image Segmentation	5
HoG	4
CV	3
SLIC	3
LBP	2
Cluster	2
MG Filter	1
Hough Transformation	1
Classify	1
Image Tiling	1
MRF	1
Masking	1
Morphological Erosion	1
Histogram Equalization	1

**Table 7 sensors-23-03670-t007:** Classification of Weeds and Crops wrt Algorithms.

Weed Type	Crop Type	DL Algorithms	Best Accuracy
Pigweed	Sugar beet, Wheat	CNN, VGGNet, SVM, ANN	95% [SVM]
Blackgrass	Sugar beet, Wheat	CNN, FCN, Mask R CNN	97% [CNN]
Bluegrass	Corn	ResNet, YOLO	99% [ResNet, YOLO]
Dockleaf	Pasture, Soybean, Turfgrass, Bahigrass	CNN, VGGNet, GoogleNet, AlexNet, DetectNet, CaffeNet	99% [AlexNet, CaffeNet]
Canadian Thistle	Corn, Cereal crops	ResNet, YOLO, VGGNet, AlexNet, GCN, SVM, ANN, DT, KNN	99% [YOLO]
Chickweed	Cranesbill, Sugar beet, Maize, Wheat, Charlock	CNN, VGGNet, FCN, Mask R CNN	95% [Mask R CNN]
Cleaver	Maize, Sugar beet, Wheat	FCN, Mask R CNN	95% [Mask R CNN]
Cockleblur	Maize, Wheat, Peanut	AlexNet	99% [AlexNet]
Crowfoot	Maize, Rice	GoogleNet	98% [GoogleNet]
Fat-Hen	Wheat, Maize, Peanut, Corn, Sugar beet	ResNet, VGGNet, FCN, YOLO, AlexNet, Mask R CNN, GCN	99% [YOLO]
Field Pansy	Wheat	CNN, VGGNet	84% [CNN, VGGNet]
Hare Ear’s Mustard	Sugar beet	SVM, ANN	95% [SVM]
Japanses Hop	Maize, Wheat, Peanut	AlexNet	99% [AlexNet]
Jungle Rice	Rice, Maize	GoogleNet, SSD	98% [GoogleNet]
Little Seed	Rice, Maize	GoogleNet, SSD	98% [GoogleNet]
Mayweed	Wheat, Maize, Carrot, Sugar beet	CNN, VGGNet, FCN, Mask R CNN	95% [Mask R CNN]
Meadow Grass	Wheat	CNN, VGGNet	84% [CNN, VGGNet]
Nutsedge	Chrysanthemum, Corn	CNN, ResNet, VGGNet, AlexNet, GCN	98% [CNN, GCN]
Paragrass	Chrysanthemum	CNN VGGNet, AlexNet, GCN	98% [CNN, GCN]
Shepherd’s Purse	Maize, Sugar beet, Wheat	SegNet, FCN, Mask R CNN	95% [Mask R CNN]
Silky-bent	Cranesbill, Sugar beet, Maize, Wheat, Charlock	FCN, Mask R CNN	95% [Mask R CNN]
Turnip Weed	Sugar beet	SVM, ANN	95% [SVM]
Dicot	Sugar beet, Soybean	FCN	87% [FCN]
Grass Weed	Wheat, Maize, Sugar beet	FCN	87% [FCN]
Velvetleaf	Tomato, Cotton	ResNet, VGGNet, MobileNet, DenseNet, Xception, SVM	99% [SVM, DenseNet]
Benghal Dayflower	Rice	SSD	86% [SSD]
Black Nightshade	Tomato, Cotton	ResNet, VGGNet, MobileNet, DenseNet, Xception, SVM	99% [SVM, DenseNet]
Hedge Bindweed	Sugar Beet	YOLO	89% [YOLO]
Indian Jointvetch	Rice, Maize	SSD	86% [SSD]
SnakeWeed	N/M	CNN	98% [CNN]
Chinee Apple	N/M	CNN	98% [CNN]
Lantana Camara	N/M	CNN	98% [CNN]
Cocklebur, foxtail, redroot pigweed and giant ragweed	Corn, Soybean	VGG16, ResNet50, Inception30, YOLOv3	98% [VGG16]
N/M	Soybean	SSD, Faster RCNN, CNN	65% [RCNN]
N/M	Spinach, beet, Bean	CNN	93% [CNN]
Dandelion, ground ivy, spotted spurge.	N/M	DCNN	92% [DCNN]
N/M	N/M	VGG16, ResNet50, DenseNet	91% [ensemble]
Cereal crops.	Monocotyledon and dicotyledon	YOLO	83% [YOLO]
Vegetables	N/M	YOLO-v3, CenterNet, and Faster R-CNN	97% [YOLO-v3]
Sugar beet	N/M	U-Net, ResNet	96% [U-net, ResNet]

## Data Availability

Not applicable.

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
