# Peer review of "Weed Detection Using Deep Learning: A Systematic Literature Review"

_sensors, 2023, doi:10.3390/s23073670_

Round 1

Reviewer 1 Report

In this work, the authors offer a systematic literature review on current state-of-the-art DL based approaches for weed detection. The manuscript is well presented and the effort made by the authors are appreciable.

The size of Table 2 and Table 3 are very large and out of page. Hence, it has to be reduced to fit completely for the page. The motivation behind the review, gaps identified in the review are to be highlighted as separate sections or subsections.

The term 'Paper' used in the headings of subsubsections 7.1.1 to 7.1.5 are to be replaced. The "Papers" is not matching in the heading. Hence, use some standard words in the heading.

The text inside Figure 11 are not readable.

The resolution of Figure 13 is not enough. ALso, few of the the text in this figure are not clear enough to read properly.

The author can use infographics on suitable places to make the manuscript more attractive. 

Author Response

First of all, we would like to thank the reviewer for the invaluable feedback. We have tried to implement the suggestions mentioned by the reviewers according to the comments by Reviewer 1 and Reviewer 2. 
The details responses can be found in the attached file.

Reviewer 2 Report

Weed Detection using Deep Learning: A Systematic Literature Review

This overly long paper does the survey of literature review (SLR) and presents details to respond to the following four questions all related to the detection and removal of weeds from the crop vegetation.

RQ1: What is the trend of employing deep learning to address the problem of weed detection in recent years?

RQ2: Which types of weeds and corresponding crops have been detected using deep learning, and what are the characteristics of the corresponding weed datasets?

RQ3: Which deep learning algorithms are best suited for a particular weed/crop combination?

RQ4: What are the tangible future research directions to achieve further benefit from deep learning applications for weed detection?

To facilitate readers, it also creates a citation graph to understand the patterns of publications and researchers in weed detections and removal. By reviewing the published literature, the authors also rate which techniques are explained best in this process.

This review of literature will help readers, program developers, and grant proposal writers to get funding to develop devices that can detect and help remove weeds at various phases of development from multiple canopied crops where light penetration and weed development vary in a complex manner. Weed detection and removal seem to be a very complicated and challenging process. To simplify this complex process, the authors have surveyed literature published since 2015 and categorized them into different categories that have developed complex algorithms to help detect weeds by color and vegetation morphologies at various stages of weed development. The authors have presented various samples of weeds with pictures. The authors also bring up various issues of training samples and present descriptions that compare such training samples with standard practices while testing the effectiveness of various algorithms. Though the development of weed detecting and removing devices is still in the testing phase, this work can be considered as one the novel ones.  This research holds great importance from scientific research, economic, and environmental perspectives. Weeds are one of the most harmful agricultural pests that negatively impact crop production, and often lead to crop failures. Devising a technique through artificial intelligence, machine learning, and deep learning techniques to detect and remove weeds will be a pioneering work and deserves publication. However, this paper is way too long to read and grasp the concepts.  It MUST be shortened in all areas.

 There are several areas that need to be rewritten and run-on sentences must be corrected. Many punctuations are missing, and there are errors in sentences, for example, lines 419-420. 

Author Response

First of all, we would like to thank the reviewers for their valuable comments and suggestions. We have tried to implement the suggestions. The detail responses to each comment are attached as a separate file.
